# Cloud ice caused by atmospheric mineral dust - Part 1: Parameterization of ice nuclei concentration in the NMME-DREAM model

Slobodan Nickovic[1], Bojan Cvetkovic[1], Fabio Madonna[2], Marco Rosoldi[2], Goran Pejanovic[1], Slavko Petkovic[1]and Jugoslav Nikolic[1]

[1]Republic Hydrometeorological Service of Serbia, 11000 Belgrade, Serbia
[2]Consiglio Nazionale delle Ricerche, Istituto di Metodologie per l'Analisi Ambientale, 85050, Tito Scalo, Potenza, Italy

*Correspondence to*:
Dr. Slobodan Nickovic
Republic Hydrometeorological Service of Serbia
11000 Belgrade, Serbia
Phone +381-65-654-8154
E-mail: nickovic@gmail.com

**Keywords**

ice nucleation, model parameterization, dust aerosol

**Abstract**

Dust aerosols are very efficient ice nuclei, important for heterogeneous cloud glaciation even in regions distant from desert sources. A new generation of ice nucleation parameterizations, including dust as ice nucleation agent, opens the way towards a more accurate treatment of cold cloud formation in atmospheric models. Using such parameterizations, we have developed a regional dust-atmospheric modelling system capable of predicting, in real-time, dust-induced ice nucleation. We executed the model with the added ice nucleation component over the Mediterranean region, exposed to moderate Saharan dust transport, over two periods lasting 15 and 9 days respectively. The model results were compared against satellite and ground-based cloud-ice-related measurements, provided by SEVIRI (Spinning Enhanced Visible and InfraRed Imager) and by the CNR-IMAA Atmospheric Observatory CIAO in Potenza, South Italy. The predicted ice nuclei concentration showed a reasonable level of agreement when compared against the observed spatial and temporal patterns of cloud ice water. The developed methodology permits the use of ice nuclei as input into the cloud microphysics schemes of atmospheric models, expecting that this approach could improve the predictions of cloud formation and associated precipitation.

# 1 Introduction

Aerosols acting as ice nucleating particles enhance the heterogeneous glaciation of cloud water making it freeze earlier and at higher temperatures than otherwise. Insoluble particles, such as dust and biological particles, are known as the best ice nuclei. Cziczo et al (2013), hereinafter referred to as CZ13, show that mineral dust and metallic oxide particles, found as residues in the ice crystals of aircraft measurements over North and Central America, are prevailing (61%). Concerning other aerosol types, CZ13 show that in the regions distant from dust sources sea salt is represented with only 3% in the regions away from the open ocean, whereas elemental carbon and biological particles represent less than 1%. Furthermore, CZ13 demonstrate that the dominant ice nucleation (IN) is heterogeneous immersion process in 94% of the collected samples. During IN, only a small number of dust particles, a few in a standard litre, are sufficient to trigger the cloud glaciation process at temperatures lower than -20°C (DeMott et al, 2015). Since dust in small concentrations is easily lifted to the mid- and upper troposphere, the cold clouds formed due to dust can be found at locations distant from dust deserts (Creamean et al, 2013; CZ13).

Mineral dust particles as significant contributors in IN are associated with the cloud formation and precipitation (Sassen, 2005; DeMott et al., 2003; Yakobi-Hancock et al, 2013). For example, the measurement of ice residues from in-situ cold cloud samples and the precipitation measurements collected in California strongly suggest that the non-soluble aerosol originating from the Asian and Saharan dust sources (dust and biological aerosol) enhances both ice formation in mid-level clouds and precipitation (Ault et al, 2011; Creamean et al, 2013). Recent modelling experiments confirm that in the pristine environment dust and biological aerosols could increase the precipitation as well (Fan et al, 2014). In this process, there is little influence of dust chemical aging (DeMott et al, 2015). Another study indicates that desert dust has the ability to glaciate the top of developing convective clouds, creating ice precipitation instead of suppressing warm rain; also dust invigoration effect would enhance precipitation ( Rosenfeld et al., 2008). On the other hand, Teller et al. (2012) conclude from their modelling study that the presence of mineral dust had a much smaller effect on the total precipitation than on its spatial distribution, which indicates that quantification of dust effects to precipitation is still uncertain because dust could modify cloud properties in many complex ways (Huang et al., 2014); therefore impacts of dust on cloud processes requires further research.

The large interest in ice nucleation research, illustrated by the exponential growth of published articles in this field (DeMott et al, 2011) has been motivated, *inter alia*, by the needs of the community to improve the unsatisfactory representation of cloud formation in atmospheric models, and therefore to increase the accuracy of weather and climate predictions. Older parameterizations (Fletcher, 1962; Meyers et al, 1992) considered the concentration of ice nuclei concentration ($n_{IN}$) only as a function of the temperature and ice saturation ratio. More recent observations, however, show that at a given temperature and moisture $n_{IN}$ depends on aerosol concentration as well. Based on this evidence, a new generation of $n_{IN}$ parameterizations has been developed (DeMott et al, 2010a; Niemand et al, 2012; Tobo et al., 2013;

Phillips et al., 2013; Atkinson at al., 2013; DeMott et al, 2015), where dust is recognized as one of the major $n_{IN}$ input parameter, in which $n_{IN}$ represents the fraction of dust aerosol capable to produce cloud water ice.

Exploiting these findings, we have developed a coupled regional real-time forecasting system (composed of the atmospheric NMME model and DREAM dust model), which predicts dust-caused $n_{IN}$ as an online model variable. When parameterizating $n_{IN}$, the immersion and deposition modes of freezing have been assumed to drive the ice formation process. Such new parameter will be used in our future study as an input to a microphysics scheme, expecting improve the operational prediction of cold clouds and associated precipitation. Currently, $n_{IN}$ is not used as online prognostic variable in eider of the operational dust models of two largest international dust forecasting networks: in the WMO Sand and Dust Warning and Assessment System (SDS-WAS) (://www.wmo.int/pages/prog/arep/wwrp/new/Sand_and_Dust_Storm.html) and in the ICAP Multi-Model Ensemble (ICAP-MME) (http://icap.atmos.und.edu/ ). Unlike dust models of these networks, our modelling system predicts $n_{IN}$ at every model time step which will be used as input to the microphysics scheme in the study of our forthcoming paper.

The model description and the implemented $n_{IN}$ parameterizations are presented in Section 2. The observations used for the model evaluation and the model performance are presented in Section 3. The comparisons of model simulations against observations are described in Section 4. Conclusions are given in Section 5.

## 2 Modelling

The evidence on the dominant role of dust in cold cloud formation has motivated a number of research groups to link cloud microphysics schemes with the parameterizations of dust-affected $n_{IN}$ in atmospheric models. The atmospheric models which drive ice nucleation parameterizations are ranging from simplified 1-D and 1.5-D kinematic or trajectory models (Field et al.,2012; Eidhammer et al., 2010; Dearden et al., 2012; Simmel et al., 2015), to complex full atmospheric models (e.g. Niemand et al, 2012; Thompson and Eidhammer, 2014). However, only a few such models (used in weather and/or climate applications) have dust concentration as a forecasting parameter available for online $n_{IN}$ calculation. For example, in a dust event study Niemand et al. (2012)  used the temperature and dust particle surface area predicted by the regional-scale online coupled model COSMOART (Consortium for Small-Scale Modelling–Aerosols and Reactive Trace Gases) to calculate immersion freezing $n_{IN}$. The model has been compared against observations using the chamber-processed $n_{IN}$ calculated from the ground-based aerosol concentration measurements, but not against directly observed cloud ice. Furthermore, Hande et al. (2015) have implemented the COSMO model coupled to the MUlti-Scale Chemistry Aerosol Transport (MUSCAT) model to compute a seasonal variability of $n_{IN}$. However, this model has not been compared against daily observations. The model which gets close to the real-time forecasting of glaciated clouds is a 'dust friendly' version of the bulk microphysics scheme (Thompson and Eidhammer, 2014), with explicitly incorporated dust aerosols. However, this model currently uses a climatological rather than predicted dust concentration for $n_{IN}$ calculations.

Following the objective of this study to develop a method for real-time $n_{IN}$ prediction, we have used the Dust Regional Atmospheric Model (DREAM) driven by the National Centers for Environmental Predictions (NCEP) Nonhydrostatic Multiscale atmospheric Model on the E grid (NMME) in which we have incorporated a parameterization of the ice nuclei concentration calculated at every model time step as a function of dust concentration and atmospheric variables.

## 2.1 NMME model

NMME (Janjic et al., 2001, 2010; Janjic, 2003) has been used for various applications at NCEP and elsewhere since the early 2000s. From 2006 it has been the main operational short-range weather forecasting North American Model (NAM). It is also used for operational regional forecasts in the Republic Hydrometeorological Service of Serbia. The NMME dynamics core includes: energy/enstrophy horizontal advection; vertical advection; a nonhydrostatic add-on module; lateral diffusion; horizontal divergence damping; sub-grid gravity waves; transport of moisture and different passive tracers. Concerning the model physics, there are various optional modules: cloud microphysical schemes ranging from simplified ones suitable for mesoscale modelling to sophisticated mixed-phase physics for cloud resolving models; cumulus parameterizations; surface physics; planetary boundary layer and free atmosphere turbulence; and atmospheric longwave and shortwave radiation schemes. NMME uses a hybrid vertical coordinate with a terrain-following sigma in the lower atmosphere, and a pressure coordinate in the upper atmosphere.

## 2.2 DREAM model

DREAM (Nickovic et al, 2001; Nickovic,, 2005; Pejanovic et al, 2011) has been developed to predict the atmospheric dust process, including dust emission, dust horizontal and vertical turbulent mixing, long-range transport and dust deposition. Eight radii bins in the model range from 0.15μm to 7.1μm. Dust emission parameterization includes a viscous sub-layer between the surface and the lowest model layer (Janjic, 1994) in order to parameterize the turbulent vertical transfer of dust into the lowest model layer following different turbulent regimes (laminar, transient and turbulent mixing). The wet dust removal is proportional to the rainfall rate. The specification of dust sources is based on the mapping of the areas that are dust productive under favourable weather conditions. The USGS land cover data combined with the preferential dust sources of dust originating from the sediments in paleo-lake and riverine beds (Ginoux et al., 2001) have been used to define barren and arid soils as dust-productive areas.

## 2.3 Ice nucleation parameterization

In this study, dust concentration, atmospheric temperature and moisture as predicted by the atmospheric component of the coupled model were used for $n_{IN}$ calculation . The $n_{IN}$ parameterization consists of two parts, applied to warmer and colder glaciated clouds.

For temperatures ranging in the interval (-36°C; -5°C), we have implemented the immersion ice nucleation parameterization developed by DeMott et al. (2015):

$$n_{IN} = C(n_{dust})^{(\alpha(273.16-T)+\beta)} exp^{(\gamma(273.16-T)+\delta)} \tag{1}$$

where $n_{IN}$ is the number concentration of ice nuclei $[l^{-1}]$; $n_{dust}$ is the number concentration of dust particles with a diameter larger than 0.5µm $[cm^{-3}]$; T is the temperature in Celsius degrees; α = 0; β = 1.25; γ = 0.46; and $d = -11.6$. Equation (1) is applied when relative humidity with respect to ice is exceeding 100%. This parameterization scheme has been developed as an extension of DeMott et al. (2010) and Tobo et al. (2013), but applied exclusively to mineral dust $n_{IN}$ collected in laboratory and field measurements. With the DeMott et al approach, the spread of errors in predicting IN concentrations at a given temperature has been reduced from the factor of ~1000 to ~10 (DeMott et al, 1010). Their parameterization is based on the use of observations from a number of field experiments at a variety of geographic locations over a period longer than a decade, demonstrating that there is a correlation between the observed $n_{IN}$ and the dust number concentrations of particles larger than 0.25µm in radius. In DeMott et al. (2015), C = 3 is chosen as a calibration factor to adjust the scheme to dust measurements. Despite the fact that validity of the scheme is for temperatures colder than -20°C we extrapolated its application down to -5°C in order to test the model if it can predict the occurrence of lower mixed clouds for the temperatures range being out of the validity of the parameterization scheme.

For temperatures in the interval (-55°C; -36°C), we have implemented the Steinke et al. (2015) parameterization for the deposition ice nucleation based on the ice nucleation active surface site approach in which $n_{IN}$ is a function of temperature, humidity and the aerosol surface area concentration. In the deposition nucleation, water vapour is directly transformed into ice at the particle's surface, at the time of or shortly after the water condensation on the particle, which acts at the same time as a condensation and freezing nucleus. Steinke et al. (2015) calculate the number concentration of ice nuclei due to deposition freezing as:

$$n_{IN} = pS_{dust} \exp[-q(T - 273.16) + (rRH_{ice}\text{-100})] \tag{2}$$

here, $n_{IN}$ is the number concentration of ice nuclei $[cm^{-3}]$; $S_{dust}$ is the the ice-active surface site density $[m^{-2}]$ (Niemand et al., 2012) describing the efficiency of a dust particle to freeze the cloud water. p = $188 \times 10^5$; $q = -1.0815$; $r = -0.815$; $T$ is temperature in degrees Celsius; $RH_{ice}$ is relative humidity with respect to ice. In our experiments, $RH_{ice}$ is prespecified to the value of 110%.

Although based on two different parameterizations, the resulting $n_{IN}$ has a smooth transition across the temperature boundary of -36°C between DeMott et al. (2015) and Steinke et al. (2015) schemes. At this transitional temperature, we have not applied any mathematical smoothing.

The schemes of DeMott et al. and Steinke et al. require temperature, relative humidity and dust concentration as input parameters, but not vertical velocity as used in some other microphysical schemes (e.g. Wang et al, 2014) .

## 3 Observations

The model capabilities to predict vertical features of dust and cold clouds have been evaluated using vertical profiles of the aerosol and cloud properties routinely measured at the CNR-IMAA Atmospheric Observatory (CIAO) at Tito Scalo (Potenza), Italy, using several ground-based remote sensing techniques, such as lidar, radar and passive techniques.

MUSA (Multiwavelength System for Aerosol) is a mobile multi-wavelength lidar system based on a Nd:YAG laser equipped with second and third harmonic generators and on a Cassegrain telescope with a primary mirror of 300 mm diameter. The three laser beams at 1064, 532 and 355nm are simultaneously and coaxially transmitted into the atmosphere in biaxial configuration. The receiving system has 3 channels for the detection of the radiation elastically backscattered from the atmosphere and 2 channels for the detection of the Raman radiation backscattered by the atmospheric N2 molecules at 607

and 387 nm. The elastic channel at 532 nm is split into parallel and perpendicular polarization components by means of a polarizer beamsplitter cube. The calibration of depolarization channels is made automatically using the ±45 method. The typical vertical resolution of the raw profiles is 3.75m with a temporal resolution of 1 min. It is worth to stress that multi-wavelength Raman lidar measurements allow the user not only to monitor the dynamical evolution of aerosol particles in the troposphere, but also to identify the different aerosol types (Burton et al., 2013; Groß et al., 2015) taking advantage of the

large number of optical properties they are able to provide, i.e. lidar ratio at two wavelengths, the Angstrom exponent, the backscatter-related Angstrom exponent, and linear particle depolarization ratio. This aerosol typing capability allows the user to classify the aerosol type acting $n_{IN}$, and especially to separate mineral dust from other types of aerosol.

CIAO, as one of the Cloudnet stations (www.cloud-net.org), applies the Cloudnet retrieval scheme to provide vertical profiles of cloud types. Cloudnet processing is based on the use of ceilometer, microwave radiometer and cloud

radar observations. For the CIAO station (Madonna et al., 2010; Madonna et al., 2011), the Cloudnet processing involves observations provided by the VAISALA CT25k ceilometer, the Radiometrics MP3014 microwave profiler, and the METEK millimetre-wavelength Doppler and polarimetric cloud radar MIRA36. In particular, MIRA36 It is a mono static magnetron-based pulsed Ka-Band Doppler radar for unattended long term observation of clouds properties. In the configuration operative at CIAO, linear polarized signal is transmitted while co- and cross polarized signals are received simultaneously to

detect Doppler spectra of the reflectivity and Linear Depolarization Ratio (LDR). The reflectivity is used to determine the density of cloud constituents while LDR helps to identify the target type. The radar has a 1 m diameter antenna and emits the microwave radiation at 35.5 GHz with a peak power of 30 kW, a pulse width of 200 ns and a pulse repetition rate of 5 KHz. The antenna beam width is 0.6° x 0.6° (gain 49 dBi) and the radar sensitivity is -40.3 dBZ at 5 km (0.1 sec time resolution) while the Doppler velocity resolution is 0.02 m/s. The linear depolarization ratio (LDR) accuracy is within +/- 2.0 dB. The

receiver calibration is within an accuracy of less than +/- 1 dB. This system is able to provide high accurate measurements of the reflectivity factor with a vertical resolution up to 15 m, though the current configuration is set to a vertical resolution of 30 m. The radar is a 3D scanning system, but Cloudnet processing makes use of zenith pointing observations only.

Cloudnet processing provides the categorization of the observed vertical profiles of cloud water categories, such as liquid droplets, ice particles, aerosols and insects. This categorization is essentially based on different sensitivities of the lidar and radar to different particle size ranges. For layers identified as ice clouds, the ice water content (with the related uncertainty) is derived from radar reflectivity factor and air temperature using an empirical formula based on dedicated aircraft measurements (Hogan et al., 2005). Consistency between Cloudnet products and Raman lidar observations of clouds performed at CIAO has also been examined (Rosoldi et al., 2016).

To complement the Potenza in-situ profiling observations and to examine how the model predicts horizontal distribution of cold clouds, the MSG/SEVIRI ice water path satellite observations were used. SEVIRI (the Spinning Enhanced Visible and InfraRed Imager), as a geostationary passive imager, is on board of the Meteosat Second Generation (MSG) systems. The high SEVIRI spatial and temporal resolution (~4km and 15min, respectively), among other advantages, provides high-quality products. The inputs to the retrieval schemes were inter-calibrated effective radiances of Meteosat-8 and 9. In our study, the daily averages of the retrieved ice water path of the SEVIRI cloud property dataset (CLAAS) were used (Stengel et al., 2013a; Stengel et al., 2013b) to compare the model results against there observations:

$$IWP = \frac{2}{3} r_I r_{eff} \tau$$

Here, IWP $[gm^{-2}]$ is the ice water path, $\tau$ is the vertically integrated cloud optical thickness at 0.6μm derived in satellite pixels assigned to be cloud filled; $r_{eff}$ is the surface-area-weighted radius of cloud particles [μm]; $r_I = 0.93 g cm^{-3}$ is the ice water density.

## 4 Model experiments and validation

The model domain covers Northern Africa, Southern Europe and the Mediterranean. The model resolution has been set to 25km in the horizontal, and to 28 layers in the vertical ranging from the surface to 100hPa. At the horizontal model resolution (which relates to the hydrostatic type of thermodynamics), clouds are resolved by the following schemes: the parameterization of grid-scale clouds and microphysics (Ferrier et al., 2002); and the parameterization of convection clouds (Janjić, 1994, 2000). The initial and boundary atmospheric conditions for the NMME model have been updated every 24 hours using the ECMWF 0.5deg analysis data. The concentration was set to zero at the 'cold start' of DREAM launched 4 days before the period to be studied, thus permitting the model to be 'warmed-up', i.e. to develop a meaningful concentration field at the date considered as an effective model start. After that time point, 24-hour dust concentration forecasts from the previous-day runs have been declared as initial states for the next-day run of DREAM.

The coupled NMME-DREAM model has been run and compared against ground-based and satellite observations for two periods (1-15 May 2010, and 20-29 September 2012) during which the CIAO Potenza instruments observed an occasional occurrence of Saharan dust accompanied with a sporadic formation of mixed-phase and/or cold clouds. These periods, characterized by modest rather than major dust transport into the Mediterranean, have been intentionally chosen to find out if non-intensive dust conditions can still form cold clouds.

For the May 2010 period, a detailed day-by-day comparison of the model against SEVIRI data is shown in Figure 1. It is important to mention that during the periods 8–9 May and 13–14 May, the Eyjafjallajökull volcanic cloud was also observed in Potenza (Mona et al, 2012; Pappalardo et al, 2013), thus potentially interfering with dust. Possible influence of the existing volcanic ash on our results is discussed later in the text.

Figure 1 shows the mapped daily averages of the following variables: the model vertical dust load (DL), the model $NL = \log_{10} \int n_{IN} dz$, the MSG-SEVIRI $IWPL = \log_{10}(IWP)$, and the overlap of NL and IWPL; columns in the Figure showing these variables are marked by A, B, C, and D, respectively. From columns (A) and (B) in Figure1 one can observe a general lack of agreement between DL and NL. This difference is expected, since the cold cloud formation is dependent not only on dust but also on its complex interaction with the atmospheric thermodynamic conditions. On the other hand, a visual

inspection shows a considerable similarity between NL and the IWPL patterns (columns (B) and (C)) with respect to their shapes and locations.

The maps in column (D) show how much the normalized NL and IWPL daily averages are overlapping. Hits, misses and false alarms are represented by areas shaded in blue, green and brown colour, respectively. One can notice that the overlapping (hits) always represents the largest parts of the shown daily maps. Although not dominant, there,are certain

regions of cold clouds either observed but not predicted (misses), or predicted but not observed (false alarms). The former case should not necessarily be erroneous because it might be addressing the processes not represented by our parameterization: the clouds generated by homogeneous glaciation or the clouds made by heterogeneous freezing with aerosols other than dust.

To gain additional evidence on the matching between NL and IWPL, we used their normalized daily averages to

calculate the following statistical dichotomous (yes/no) scores based on hits, misses and correct negatives (not predicted, not observed) (WMO, 2009):

- accuracy - showing what fraction of the forecasts were correct;
- probability of detection (hit rate) - showing what fraction of the observed "yes" events were correctly forecasted;

- the false alarm ratio - showing what fraction of the predicted "yes" events actually did not occur.

The scores were calculated using the values for all model/observation grid points and for all days of the considered period. Figure 2 shows the time evolution of the scores (which by definition range between 0 and 1). In average for the whole period, 63.4% of all NL were correct with respect to IWPL, 73.9% of the observed IWPL were predicted, and for 30.4% of the forecast NL, IWPL was not observed. Such result confirms a high matching level between two fields shown in

Figure 1.

Additional evidence on matching between our forecasts and satellite observations has been made by applying the Method for Object-based Diagnostic Evaluation - MODE (Davis et al., 2006a; 2006b; 2009) which is based on a fuzzy-logic algorithm and which has been originally developed to quantify the errors related to spatial patterns and location of precipitation which considers various attributes of rain patterns (e.g. orientation, rain area). Factors as the separation of the

object (pattern) centroids, minimum edge separation between modeled and observed patterns, model/observed patterns orientation angles relative to the grid axis, the ratio of the areas of the two objects, and the fraction of area common to both objects. MODE is used here to indicate the level of matching between NL and IWPL for a selected day of 11 May 2010. Figure 3 shows that MODE has identified three precipitation objects: two (green and red colored) showing good matching, and one (blue) with no matching.

To evaluate the model performance in representing the vertical structure of the ice water clouds, $n_{IN}$ was compared with the observed IWC obtained using the Cloudnet retrieval scheme over Potenza. Figure 4 shows time evaluation of $\log_{10}(n_{IN})$ (colour shaded) and $\log_{10}(\text{IWC} \times 10^{-6}\frac{kg}{m^3})$, (contour plotted) over periods 1-15 May 2010 and 22-30 September 2012. In addition, the red contours show the temperature field as provided by the NMME model. The different quantities provided by DREAM and Cloudnet to characterize the cloud ice content. Note that IWC $n_{IN}$ are different physical variables and therefore a semi-quantitative comparison is possible.

The comparisons reveal general good performances of DREAM in predicting the vertical structure of the observed ice clouds for temperatures below -20°C which coincides with the validity range of DeMott et al. (2015). Especially during 1-15 May 2010, a remarkable agreement between patterns of the ice vertical layer retrieved using the cloud radar observation and the $n_{IN}$ predicted by the model is evident. However, most of the ice nuclei concentration for temperatures warmer than -20°C was not predicted, although mixed clouds were observed by the cloud radar below 4.0. This is particularly evident on 6 May when ice cloud layer below 3 km AGL was observed only by the radar, and conditions for ice occurrence were completely missed by the model.

On 22-30 September 2012 the model was able to indicate of the deep ice layers observed on 25-27 September 2012 between about 5 and 12 km AGL (-10°C and -60°C) and it was able to partially predict a part of the thinner layers observed after 27 September above 7 km AGL (<-25°C). The model was also able to predict well the occurrence of cirrus clouds observed by the cloud radar on 29 September in the range between 6 and 12 km. It is also worth to mention that the co-located and simultaneous Raman lidar measurements (not reported) showed some high optically thin cloudiness not detected by the radar because of its limited sensitivity to thin clouds at that height (Borg et al., 2013). In particular, the $n_{IN}$ layers predicted by the model in the second half of 27 September and on 28 September are in the range between 9 and 12 km. However, as in the case of May 2010, the model underpredicted $n_{IN}$ for the lowest ice water layers observed with the radar below 4 km.

Inability of the model to predict $n_{IN}$ at lower elevations can be explained by the fact that the DeMott et al. (2015) parameterization is valid for temperatures in the interval (-20°C – -36°C). We extended this scheme to work in the interval (-5°C ; -20°C) as well but our experiments showed that lower mixed clouds could not be predicted. This result is consistent with the statement of DeMott et al. (2015) that the parameterization is weakly constrained at temperatures warmer than -20°. As these authors also claimed, this is the temperature regime that may be dominated by organic ice nucleating particles such as ice nucleating bacteria, which is aerosol not included in our parameterizations.

In Figure 5 we also report the comparison of IWPL and NL over Potenza calculated every three hours, in the period from 1 to 15 May 2010 (left panel) and from 22 to 30 September 2012 (right panel). The outcome of the comparison confirms the good performance of the model in the prediction of $n_{IN}$ of the ice clouds over the whole atmospheric column.

The correlation between the IWPL and NL retrieved using the ground based measurements, merging the datasets from both selected cases studies of 1-15 May 2010 and 22-30 September 2012, is shown in Figure 6. The linear correlation made considering the daily averages for both quantities provides a regression coefficient of R=0.83. The scatter plot shows a large variability in the values corresponding to the higher values of the IWP and to the higher values of IL. Therefore, for optically thinner ice clouds, IL linearly increases with IWPL. For larger IWPL values, the two variables are less correlated and the second or higher order polynomial fitting could compromise the linear relationship. In the period between 13-15 May 2010, both DREAM and back trajectories analysis showed that, while the transport of volcanic aerosol from Iceland (due to the Eruption of volcano Eyjafjalla 2010) was still ongoing, dust contribution was not negligible (Mona et al., 2012). In this period, the volcanic aerosol was mainly 5 transported across the Atlantic Ocean, passing over Ireland and west UK, and then transported to the west off the Iberian Peninsula before reaching the Mediterranean Basin and Southern Italy. Satellite images and ground-based measurements confirmed the presence of volcanic particles in the corresponding regions (not shown). A detailed description of the volcanic layers as observed by EARLINET (European Aerosol Research Lidar NETwork) during this period is reported in Pappalardo et al., 2014. EARLINET volcanic dataset is freely available at www.earlinet.org (The EARLINET publishing group 2000-2010; (2014): EARLINET observations related to volcanic eruptions (2000-2010); World Data Center for Climate (WDCC). http://dx.doi.org/10.1594/WDCC/EN_VolcanicEruption_2000-2010). Moreover a devoted relational database freely available at www.earlinet.org contains all information about volcanic layers (base, top, centre of mass) and correspondingly mean and integrated values.

The Iberian Peninsula, France and South Italy were the regions more significantly affected by the presence of volcanic aerosol (sulphate and small ash) during the considered period. For the purpose of our modelling study this might induce an underestimation of the IN (since IN due to dust only is modelled) in the above mentioned regions and can be responsible of part of the discrepancies between modelled IN and IWP provided by SEVIRI. This is particularly true for Iberian Peninsula where volcanic aerosol concentrations were quite relevant. The comparison of model predicted IN and SEVIRI IWC on 13 May shows differences that might be correlated to a larger availability of IN of volcanic origin. On the contrary, in South Italy, the volcanic layer, observed at Potenza up to an altitude up to 15.8 km above sea level, did not enhance the formation of cold clouds due to unfavourable dry conditions in the free troposphere; this is also confirmed by the Potenza cloud radar which did not observed? clouds for the whole day (Figure 4). The absence of cold clouds over most of South Italy, including Potenza region, is also shown by the IWC reported for 13 May in Figure 1.

## 5 Conclusions

We have expanded the regional DREAM-NMME modelling system with the on-line parameterization of heterogeneous ice nucleation caused by mineral dust aerosol. We employed the recently developed empirical parameterizations for immersion and deposition ice nucleation that include dust concentration as a dependent variable for the cloud glaciation process. In our approach, the ice nucleation concentration was calculated as a prognostic parameter depending on dust and atmospheric thermodynamic conditions. To our knowledge, this is one of first attempts to predict in real time all ingredients needed for parameterization of dust-induced cold cloud formation within one modelling system. Experimental NMME-DREAM $n_{IN}$ daily predictions compared against SEVIRI observations are posted at http://dream.ipb.ac.rs/ice_nucleation_forecast.html to show operational capabilities of the methodology presented in this study.

The model was applied for the Mediterranean region and surroundings for two periods: 1-15 May 2010 and 22-30 September 2012 during which several dust transport events of moderate intensity occurred. The model has been compared against both ground-based and satellite observations for two periods with the aim of checking the performance over both the horizontal and vertical cross-sections of the investigated atmosphere providing promising results. Somewhat lower performance of the model in representing ice layers at lower altitudes could have been affected by the capability of the parameterization scheme to predict mixed-phase clouds in the zone of warmer negative temperatures.

Our study aimed to develop a methodology which lays the groundwork for further improvement of predicting clouds and associated precipitation in current atmospheric models. Namely, the operational numerical weather prediction systems today usually do not include aerosol effects in cloud formation or they do it in a simplistic way. By integrating dust and atmospheric components into an modelling system we achieved to have all necessary ingredients at every time step – atmospheric and aerosol parameters – to calculate the ice nuclei concentration formed by dust, which will be used in our future development as an input into a dust-friendly cloud microphysics to predict the ice mixing ratio.

**Acknowledgments**

We acknowledge the EUMETSAT for use of its Satellite Application Facility on Climate Monitoring (CM SAF) data. We are grateful to Dr Beth Ebert, Bureau of Meteorology Research Centre, Melbourne (Australia) for guidance and discussions on object-oriented model validations. We thanks to Luka Ilic, Institute of Physics Belgrade, Serbia, and to Milica Arsic, Republic Hydrometeorological Service of Serbia, for their technical assistance. The support for the modeling part of the study is provided by the Republic Hydrometeorological Service of Serbia; the support for the part of the study related to observation is provided by CNR-IMAA, EARLINET under EU grant RICA 025991 in the Sixth Framework Programme, Cloudnet project (EU contract EVK2-2000-00611) for providing the ice water content which was produced by CNR-IMAA using measurements from Potenza, and ACTRIS through EU Seventh Framework Programme (FP7/2007-2013) under grant 262254 (Including ACTRIS TNA).

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

# Figures

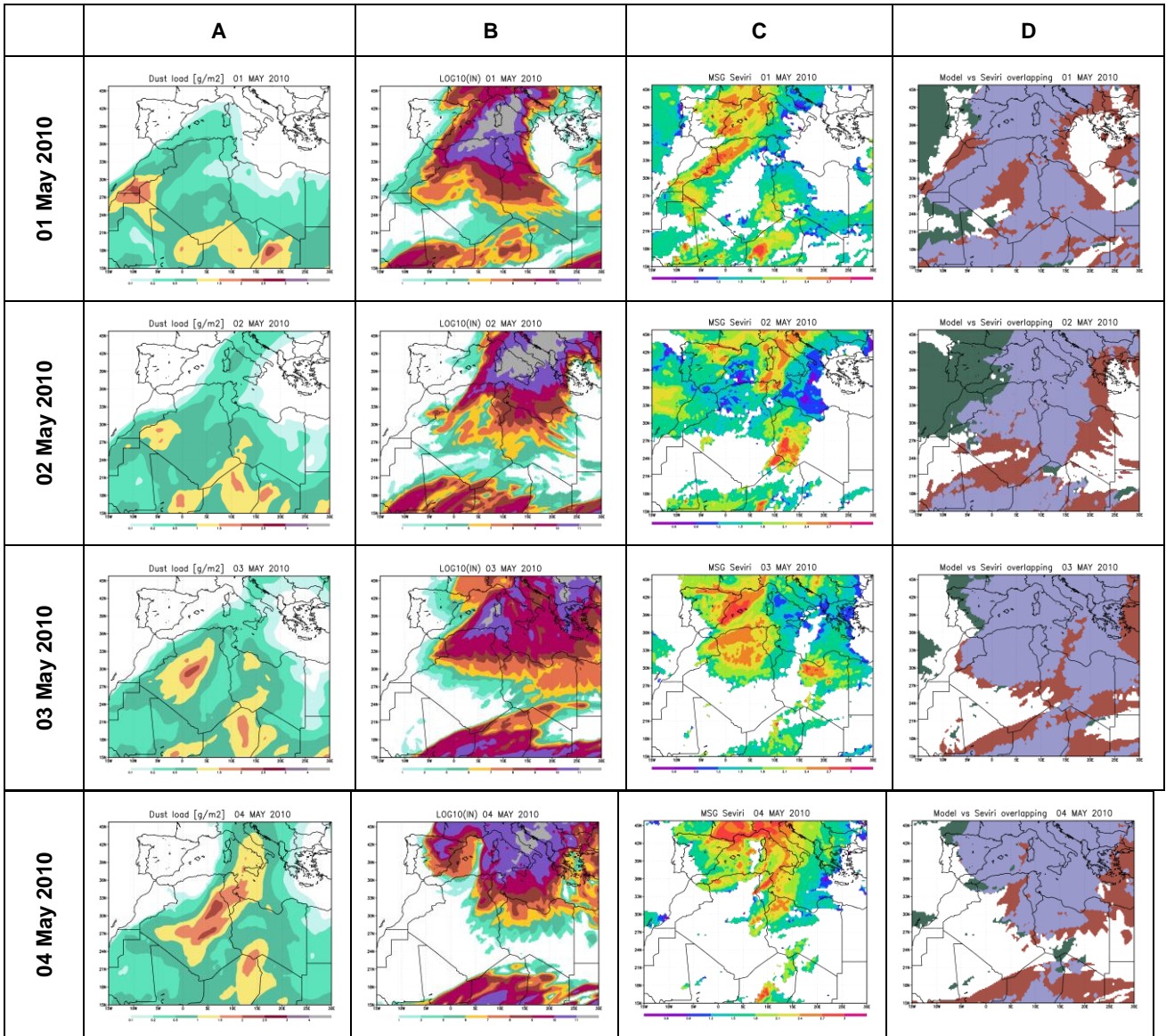

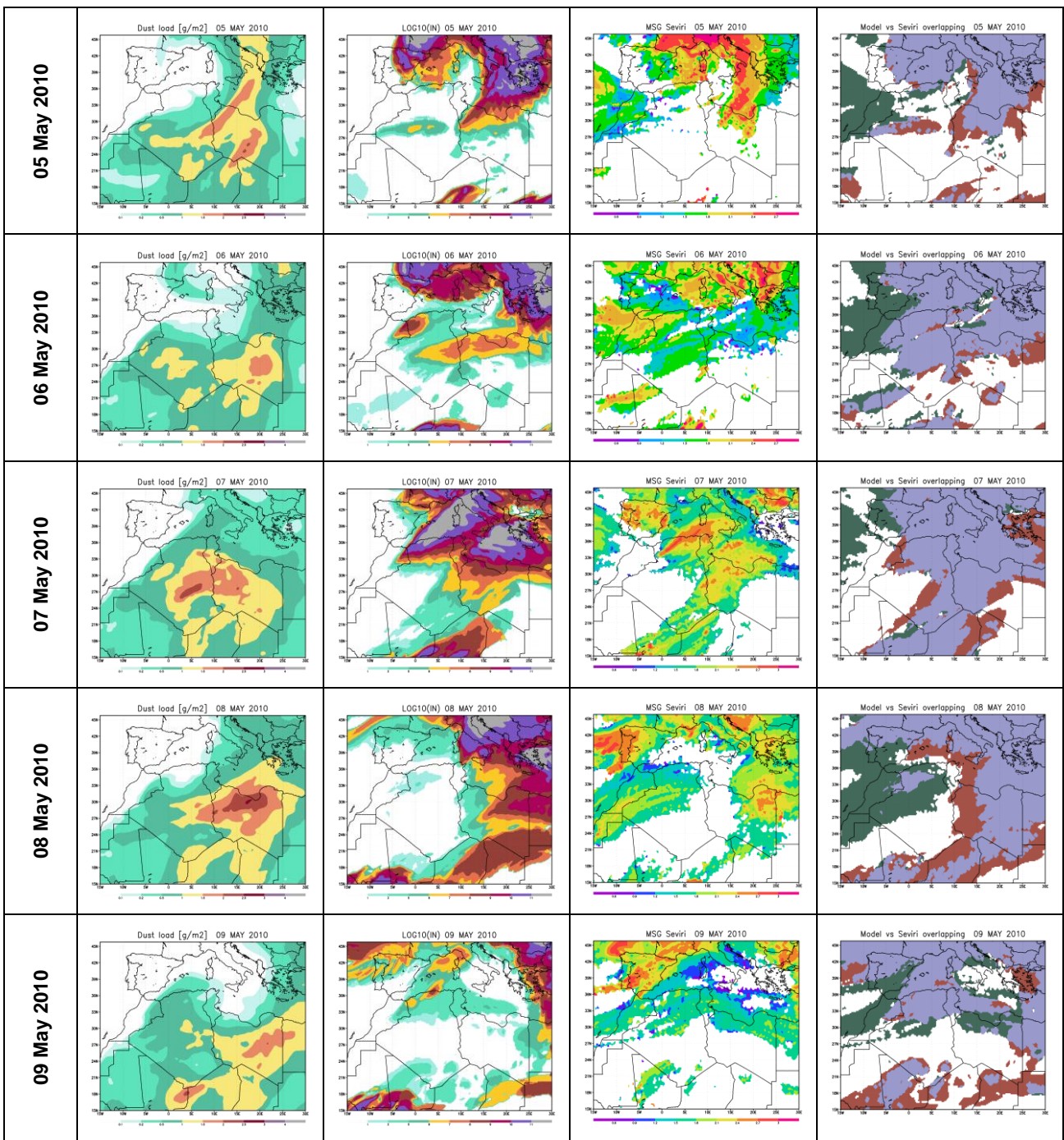

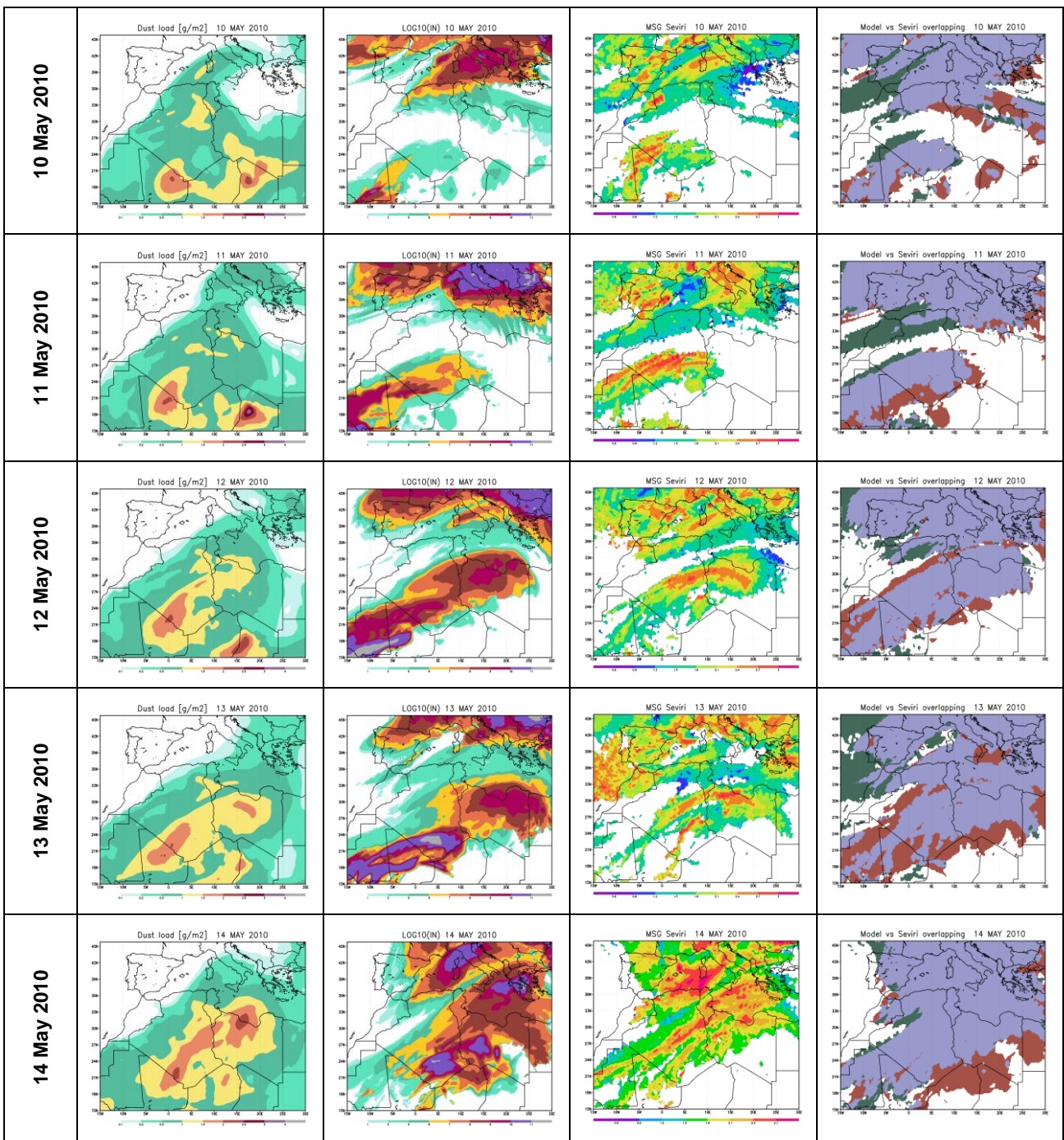

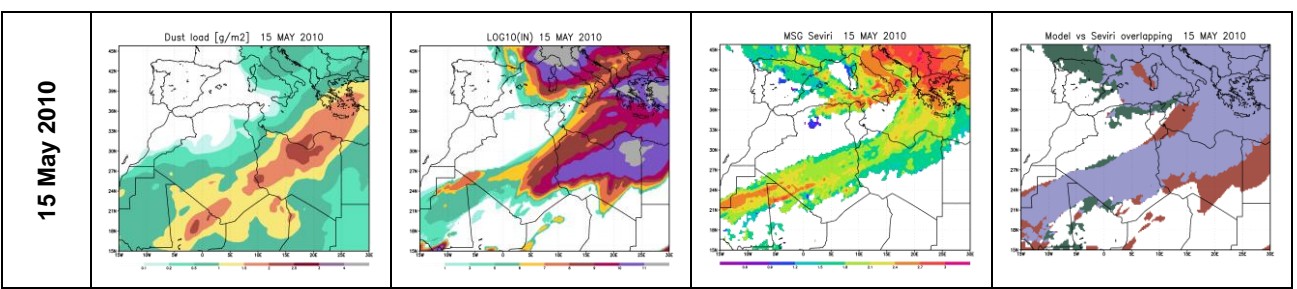

**Figure 1: Daily-averages of (A) the model dust load (gm$^{-2}$); (B) the model** $\mathrm{NL} = \log_{10} \int n_{IN} \mathrm{d}z$**; (C) the MSG-SEVIRI** IWPL=$\log_{10}(IWP)$**; (D) overlap of normalized NL and IWPL. Color selection: hits – blue; misses – green; false alarm – brown.**

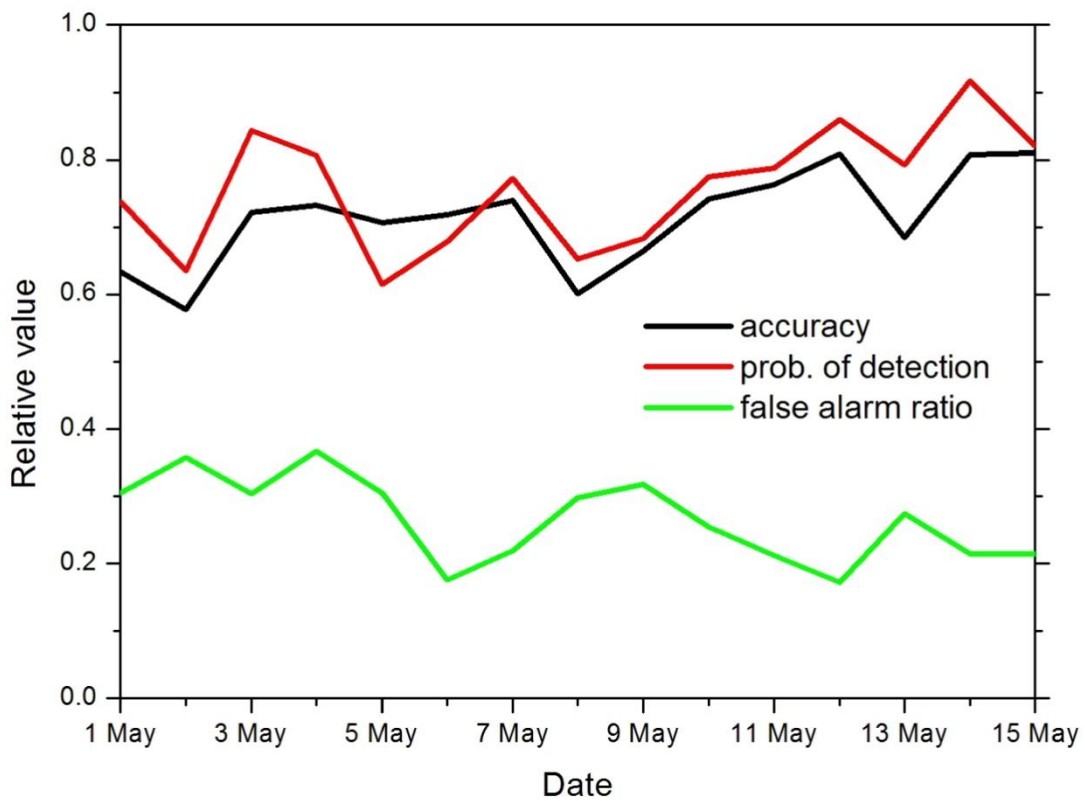

**Figure 2: Time evolution of the forecast accuracy (black), the probability of detection (hit rate, red) and the false alarm ratio (green) for the period 1-15 May 2010.**

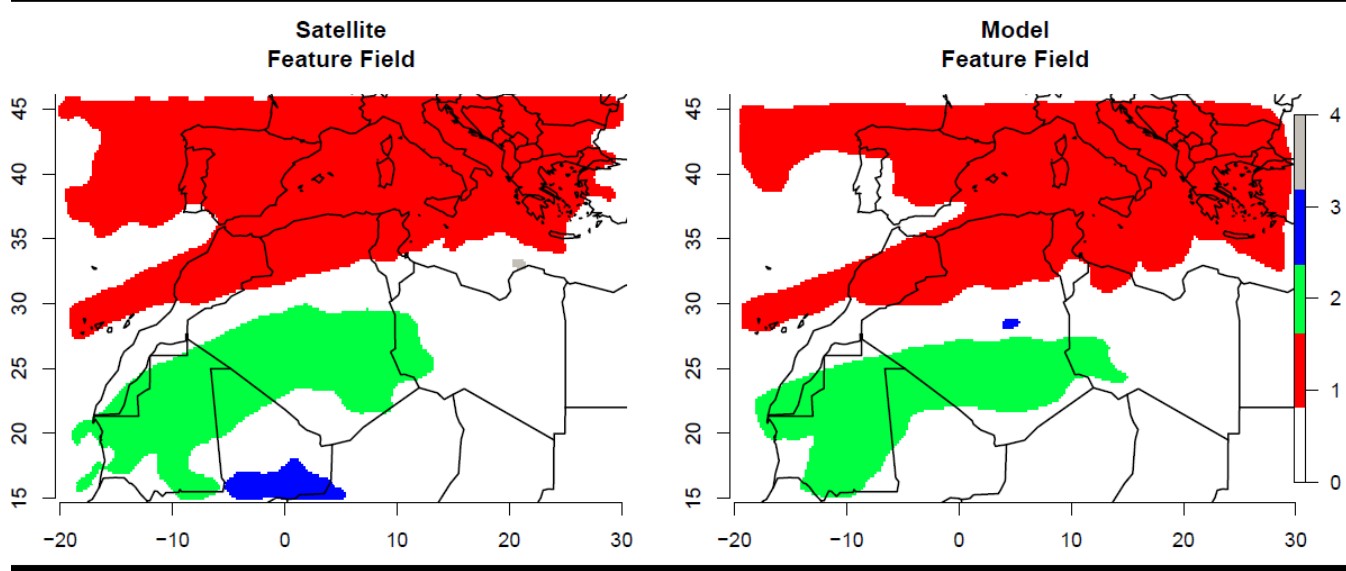

**Figure 3: Best matching pairs of entities identified using the MODE method: red and green colours identify objects that are matched, dark blue and gray indicates no matching; x-axe and y-axe are longitudes and latitudes, respectively.**

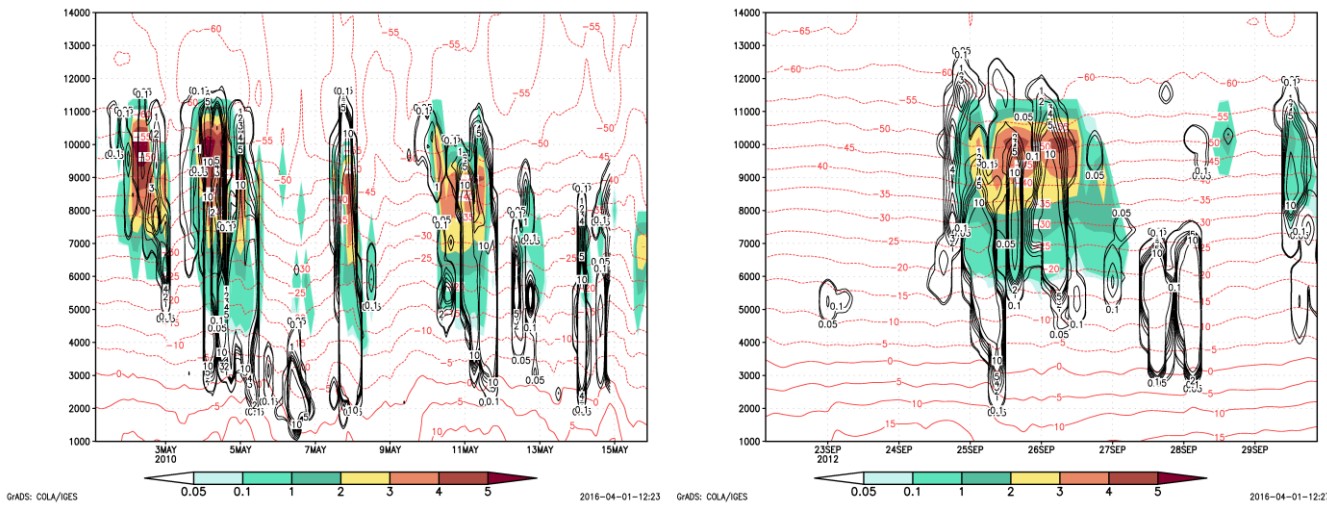

**Figure 4: Comparison of $\log_{10}$ (IWC $\times 10^{-6} \frac{kg}{m^3}$) obtained from the Doppler radar reflectivity using the Cloudnet algorithm (solid black line contour plot) versus DREAM $\log_{10}(n_{IN})$ (coloured shaded plot), in the period 1-15 May 2010 (left), and 22-30 September 2012 (right). Red contours show temperature as provided by the NMME model.**

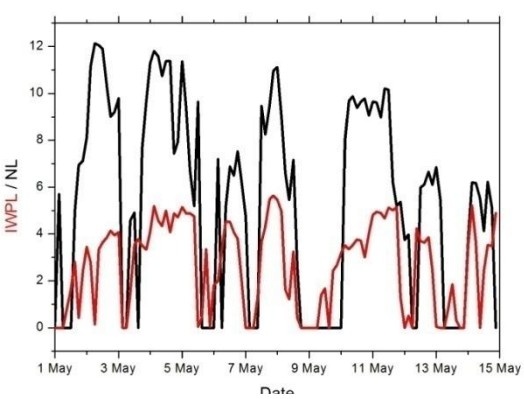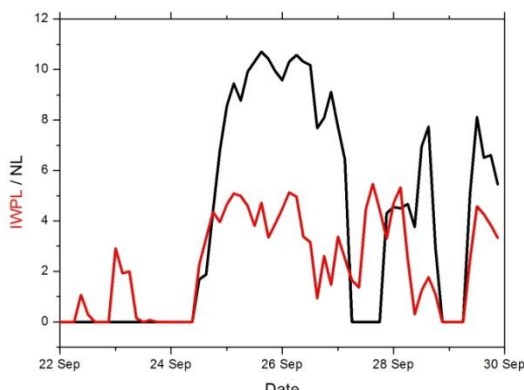

**Figure 5:** Time evolution of IWPL and NL over the periods 1-15 May 2010 (left) and 22-30 September 2012 (right).

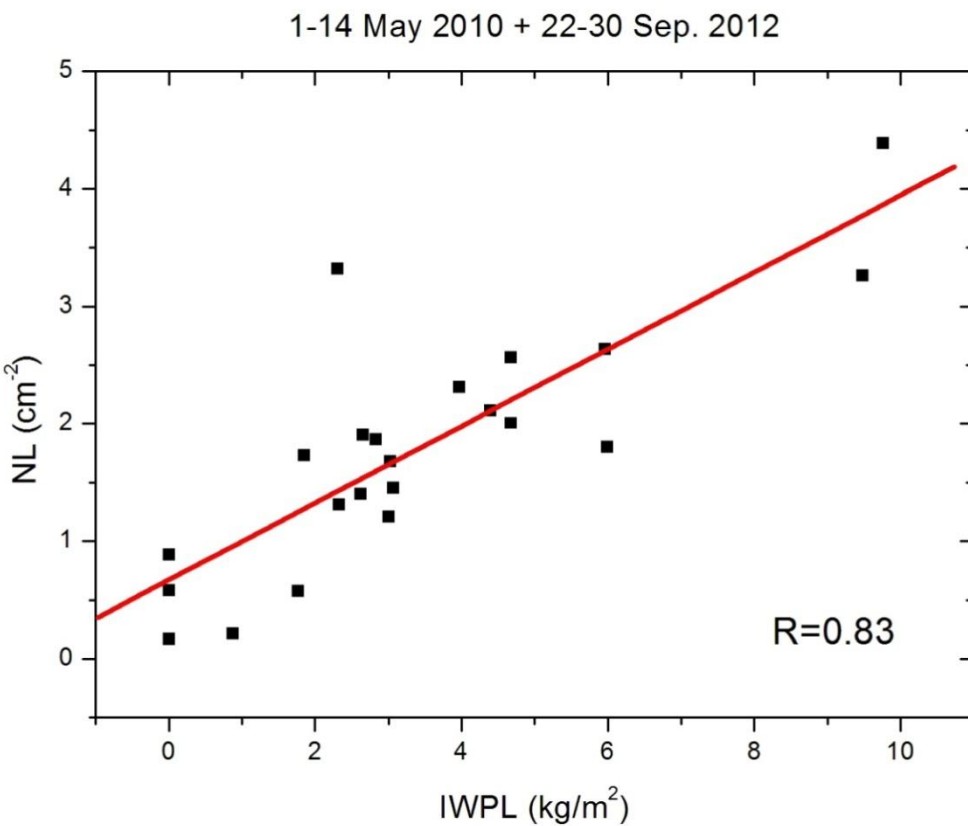

**Figure 6: Linear correlation between IWPL and NL retrieved using the ground based measurements merging the datasets from both selected case studies: of 1-15 May 2010 and 22-30 September 2012.**

