# Peer review of "Predicting cloud ice nucleation caused by atmospheric mineral dust"

_Atmospheric Chemistry and Physics, 2016_

## Referee Comment (RC1) · Anonymous Referee #1 · 10 Jun 2016

This work presents the implementation of DeMott et al., 2015 and Steinke et al., 2015 ice nuclei parameterizations in the regional atmospheric and dust transport model NMME-DREAM. Comparisons with satellite and ground measurements of cloud properties indicate a promising behavior of the new model version.

I would recommend publication in ACP after the authors address the following comments in order to clarify certain aspects of the manuscript.

Specific Comments:

In general: My main concern is that at this stage the authors compare model ice nuclei with observations of ice water path which are not directly comparable. It would be more appropriate to calculate the corresponding modeled cloud properties and use them for model evaluation. Model results with the old version of the model should be

also presented for comparison.

Page1, Line 1 (title): Predicting cloud ice nucleation caused by atmospheric mineral dust. The title could be more specific and declare that the paper shows the implementation of existing ice-nucleation parameterizations in NMME-DREAM.

Page2, Line 28: To our knowledge, this is the first time that all ingredients needed for cold cloud formation by dust are predicted in operational forecasting mode within one modeling system. Please provide more support for this statement since a variety of coupled dust-ice models seem to be already available (e.g. Zhang et al.,2012; Liu et al., 2012; Atkinson et al., 2013)

Page 4, Line 14 : In this study, dust concentration, atmospheric temperature and moisture as predicted by the atmospheric component of the coupled model are used to calculate. The parameterization consists of two parts applied to warmer and colder glaciated clouds. The vertical wind component is a crucial parameter for CCN/IN activation processes. Do you consider w in your calculations?

Page 5, Line 20: to identify the different aerosol types (Papagiannopoulos et al., 2015) taking advantage of the large number optical properties they are able to provide, i.e. lidar ratio at two wavelengths, the Angstrom exponent, the backscatter-related Angstrom exponent, and linear particle depolarization ratio. This aerosol typing capability allows to classify the aerosol type acting Nin, and especially to separate mineral dust from other types of aerosol Please add Papagiannopoulos et al., 2015 in your Reference list. Also check carefully your references and edit your list in ACP format.

Page 6, Line 15: The model resolution has been set to 25km in the horizontal. Could you please justify how you resolve cloud-scale features at this resolution?

Page 7, Line 4. On the other hand, a visual inspection shows considerable similarity between NL and the IWPL patterns (columns (B) and (C)) with respect to their shapes and locations. These two quantities are not directly comparable. Could you please

show what is the NMME predicted IWP? Also show the difference between the control run (without IN parameterization) and the new run.

Figure 2. If I interpret correct the plots in Figure 2, it seems that the model predicts IN even at areas without dust. If your only aerosol source is dust (Eq.1, Eq.2) could you please explain more on this?

Page 7, Line 29: The forecasts are translated horizontally over the observations until the minimum squared error (MSE) is achieved Please explain.

Page 8, Line 8: Anyhow, in order to predict IWC we need to incorporate predicted Nin into a cloud microphysics scheme, which is a future task of our project. Therefore, the comparison using a semi-quantitative approach is the only available at the current stage of the analysis. Why don't you incorporate the NMME microphysics scheme? Please show also the modeled IWC.

Page 8, Line 14: Most of the ice, observed by the cloud radar below 4.0-4.5 km above ground level (AGL), is not predicted by the model. Is there any dust at these layers? If there is no dust in the model and your only IN source is dust, this could make sense.

Page 8, Line 23: Moreover, like for the case of May 2010, the model tends to under-predict the lowest ice water layers observed with the radar below 4.5 km AGL. Again it is a little confusing when you refer to IN and when you refer to ice water. Also to me it looks like there is no IN below 6km which means that the model fails to represent half of the clouds in this cross-section.

Page 8, Line 27 and Figure 5 caption. Replace upper and lower panel with left and right

Page 9, Line 18: On the contrary, in South Italy, the volcanic layer, observed at Potenza up to an altitude of about 8 km above sea level Please provide some evidence for this argument

Page 9 Line 20: did not observed? Typo - observe

Page 9, Line 29 : The model has been validated Avoid the use of the term validation (here and elsewhere in the text) since you are only referring to specific case studies. A validation process would require much more comparisons with observations and for a much longer time period until the model could be verified to produce validated products.

Page 9, Line 32: wormer negative temperatures Typo - warmer

Page 10, Line 6: What do you mean by "unfired modelling system"

Throughout the text Please check again the text for grammar and spelling and provide an improved manuscript.

References

James D. Atkinson, Benjamin J. Murray, Matthew T. Woodhouse, Thomas F. Whale, Kelly J.Baustian, Kenneth S. Carslaw, Steven Dobbie, Daniel O'Sullivan & Tamsin L. Malkin, The importance of feldspar for ice nucleation by mineral dust in mixed-phase clouds, Nature 498, 355–358 doi:10.1038/nature12278, 2013

K. Zhang, D. O'Donnell, J. Kazil, P. Stier, S. Kinne, U. Lohmann, S.Ferrachat, B. Croft, J. Quaas,H. Wan, S. Rast, and J.Feichter, The global aerosol-climate model ECHAM-HAM, version 2:sensitivity to improvements in process representations, Atmos. Chem. Phys., 12, 8911–8949, 2012 www.atmos-chem-phys.net/12/8911/2012/doi:10.5194/acp-12-8911-2012

X. Liu, X. Shi, K.Zhang, E. J. Jensen, A. Gettelman, D. Barahona, A.Nenes, and P. Lawson, Sensitivity studies of dust ice nuclei effect on cirrus clouds with the Community Atmosphere Model CAM5, Atmos. Chem. Phys., 12, 12061–12079, 2012 www.atmos-chem-phys.net/12/12061/2012/ doi:10.5194/acp-12-12061-2012

---

## Referee Comment (RC2) · Anonymous Referee #2 · 13 Jun 2016

This paper develop a regional dust-atmospheric modeling system, considering dust aerosol's effect on IN, present some new and interesting results. I recommend accepted this paper for publication to do the revisions that specified below:

Page 2 line 28: To our knowledge, this is the first time that all ingredients needed for cold cloud formation by dust are predicted in operational forecasting mode within one modeling system. Please give more evidence.

Page 4 line 25: ......the spread of errors in predicting IN concentrations at a given temperature has been reduced from a factor of $\sim$1000 to $\sim$10. Please give some evidence or support for this conclusion.

Page 4 line 29: Why do you choose -5°C, since the underlying measurements were only taken at temperatures lower than -9°C. Moreover, you set the temperatures for

warmer clouds range -36âĐČ to -10âĐČ (page 4 line 17 ). Please give more discussion.

Page 5 line 9: Sdust is ice nucleation active surface linked to dust concentration. As we know that dust aerosols lifted to the mid and upper troposphere can serve as ice nuclei, here you use the surface value of dust. Will it affect the model results?

Page 5 line 12-13: Please give some related evidence.

Page 5: paragraph 1 on section 3, please give details description for the ground observe instruments.

Page 8 line 14-15: Why the model can't predict the ice below 4-4.5km while the cloud radar can detect? Due to the temperatures you set in section 2.3(-10∼–36âĐČ) , the vertical distribution of dust aerosols, or any other reason? You should give more discussion.

Page 8 line 27: The position for the pictures in Fig.5 should be left and right.

Page 9 line 20: there is a redundant question mark.

Figure 1: The colorbar and coordinate are unclear. The compared results for the second case should also be given and discussed.

Figure 3: Please give the meaning for each color and the title for x-y coordinate.

Figure 5: For the first case, the mean values of IWPL are mainly greater than NL. However, for the second case, the mean values of IWPL are mainly less than NL. Please give more discussion.

There are some research discuss dust aerosols effect on clouds and precipitation, please discuss more about the relationship between dust and clouds in Section 1.

References:

Wang, W., J. Huang, P. Minnis, Y. Hu, J. Li, Z. Huang, J. Ayers, and T. Wang, Dusty

cloud properties and radiative forcing over dust source and downwind regions derived from A-Train data during the Pacific Dust Experiment, Journal of Geophysical Research, 115 (2010), D00H35, doi:10.1029/2010JD014109.

Huang, J., P. Minnis, B. Lin, Y. Yi, S. Sun-Mack, T. Fan, and J. Ayers, 2006: Determination of ice water path in ice-over-water cloud systems using combined MODIS and AMSR-E measurements, Geophysical Research Letters, 33 (21)L21801, doi:10.1029/2006GL027038.

Huang, J., P. Minnis, B. Lin, Y. Yi, M. Khaiyer, R. Arduini, A. Fan, and G. Mace, 2005:Advanced retrievals of multilayered cloud properties using multispectral measurements, Journal of Geophysical Research, 110 (D15) (2005), D15S18, doi:10.1029/2004JD005101.

---

## Author Comment (AC1) · 24 Aug 2016

General Comment:

My main concern is that at this stage the authors compare model ice nuclei with observations of ice water path which are not directly comparable. It would be more appropriate to calculate the corresponding modeled cloud properties and use them for model evaluation. Model results with the old version of the model should be also presented for comparison.

*We agree, it would be more appropriate to compare the model against the observed cloud ice. However, in the current stage of our work, we do not predict the cloud ice but calculate the ice nuclei concentration $(n_{IN})$ and therefore compared $n_{IN}$ against the observed ice water. Although the two variables are not directly comparable as the Referee #1 correctly noticed, they are linked through the continuity equation for cloud ice and we assumed the two variables should be correlated. Our results indeed confirm that $n_{IN}$ in general well compares against the observed cloud ice water.*

*In our next paper which will represent as continuation of the current study, we will use $n_{IN}$ as a prognostic input in the cloud microphysics scheme of our dust-atmosphere modelling system. Based on above considerations, we propose to the Editor replacement of the current manuscript title with: 'Cloud ice caused by atmospheric mineral dust - Part 1: Parameterization of ice nuclei concentration in the NMME-DREAM model'. This also takes into account the Referee's suggestion in his/her first Specific Comment to change the article title. The second part of the current in that case will be a paper titled as: 'Cloud ice caused by atmospheric mineral dust - Part 2: Parameterization of ice water in the NMME-DREAM model'.*

Specific Comments:

Page2, Line 28: To our knowledge, this is the first time that all ingredients needed for cold cloud formation by dust are predicted in the operational forecasting mode within one modeling system. Please provide more support for this statement since a variety of coupled dust-ice models seem to be already available (e.g. Zhang et al.,2012; Liu et al., 2012; Atkinson et al., 2013).

*This is true there are numerous coupled dust-ice models. Articles the suggested by the Referee #1 (Zhang et al.,2012; Liu et al., 2012; Atkinson et al., 2013) are focused on studying nucleation effects in general, rather than on using dust-ice parameterization to improve numerical weather prediction. For example, $n_{IN}$ is not online prognostic variable in none of the operational dust models within the largest two international dust forecasting projects: the WMO Sand and Dust Warning and Assessment System https://www.wmo.int/pages/prog/arep/wwrp/new/Sand_and_Dust_Storm.html; and the ICAP Multi-Model Ensemble (ICAP-MME) http://icap.atmos.und.edu/ ; Sessions et al, 2015). Differently from those dust models, we perform prediction of $n_{IN}$ at every model time step to be used as input to a microphysics scheme.*

*We added in the text the following clarification:*

... Such new parameter will be used in our future study as an input to a microphysics scheme, expecting improve the operational prediction of cold clouds and associated precipitation. Currently, $n_{IN}$ is not used as online prognostic variable in eider of the operational dust models of two largest

international dust forecasting networks: in the WMO Sand and Dust Warning and Assessment System (SDS-WAS) (://www.wmo.int/pages/prog/arep/wwrp/new/Sand_and_Dust_Storm.html) and in the ICAP Multi-Model Ensemble (ICAP-MME) (http://icap.atmos.und.edu/ ). Unlike dust models of these networks, our modelling system predicts $n_{IN}$ at every model time step which will be used as input to a microphysics scheme in the study of our forthcoming paper...

Page 4, Line 14 : In this study, dust concentration, atmospheric temperature and moisture as predicted by the atmospheric component of the coupled model are used to calculate. The parameterization consists of two parts applied to warmer and colder glaciated clouds. The vertical wind component is a crucial parameter for CCN/IN activation processes. Do you consider w in your calculations?

*We wanted to implement the most recent parameterizations available in the community for dust-induced IN (DeMott et al., 2015; Steinke et al., 2015). These schemes require temperature, relative humidity and dust concentration as input parameters, but not vertical velocity. We added the following text:*

The schemes of DeMott et al. and Steinke et al. require temperature, relative humidity and dust concentration as input parameters, but not vertical velocity as it is used in some other microphysical schemes (e.g. Wang et al, 2014) .

Page 5, Line 20: to identify the different aerosol types (Papagiannopoulos et al., 2015) taking advantage of the large number optical properties they are able to provide, i.e. lidar ratio at two wavelengths, the Angstrom exponent, the backscatter-related Angstrom exponent, and linear particle depolarization ratio. This aerosol typing capability allows to classify the aerosol type acting Nin, and especially to separate mineral dust from other types of aerosol Please add Papagiannopoulos et al., 2015 in your Reference list. Also check carefully your references and edit your list in ACP format.

*The reference by Papagiannopoulos et al. has been replaced by the following two as more appropriate:*

*Groß, S., Freudenthaler, V., Schepanski, K., Toledano, C., Schäfler, A., Ansmann, A., and Weinzierl, B.: Optical properties of long-range transported Saharan dust over Barbados as measured by dual-wavelength depolarization Raman lidar measurements, Atmos. Chem. Phys., 15, 11067-11080, doi:10.5194/acp-15-11067-2015, 2015.*

*Burton, S. P., Ferrare, R. A., Vaughan, M. A., Omar, A. H., Rogers, R. R., Hostetler, C. A., and Hair, J. W.: Aerosol classification from airborne HSRL and comparisons with the CALIPSO vertical feature mask, Atmos. Meas. Tech., 6, 1397-1412, doi:10.5194/amt-6-1397-2013, 2013.*

Page 6, Line 15: The model resolution has been set to 25km in the horizontal. Could you please justify how you resolve cloud-scale features at this resolution?

*We added the following text as a clarification:*

At the horizontal model resolution used in our study (which relates to the hydrostatic type of thermodynamics), clouds are resolved by the following schemes: the parameterization of grid-scale clouds and microphysics (Ferrier et al., 2002); and the parameterization of convection clouds (Janjić, 1994, 2000).

Page 7, Line 4. On the other hand, a visual inspection shows considerable similarity between NL and the IWPL patterns (columns (B) and (C)) with respect to their shapes and locations. These two quantities are not directly comparable. Could you please show what is the NMME predicted IWP? Also show the difference between the control run (without IN parameterization) and the new run.

*See please above our reply to the General Comment of the Referee #1*

Figure 2. If I interpret correct the plots in Figure 2, it seems that the model predicts IN even at areas without dust. If your only aerosol source is dust (Eq.1, Eq.2) could you please explain more on this?

*We think the Referee #1 is reporting to Figure 1, not to Figure 2.*
*Figure 1 shows maps of dust C and IN integrated vertically (columns A and B respectively). The maps for same valid times indeed do not fully match because C and IN are not linearly proportional (Eq.1, Eq.2), neither their loads. However, according to Eq.1 and Eq.2 even for small C concentration there is some $n_{IN}$ if thermodynamic conditions permit it. With the used color palette scales in maps, load of small dust concentrations cannot be shown even if $n_{IN}$ is displayed.*

Page 7, Line 29: The forecasts are translated horizontally over the observations until the minimum squared error (MSE) is achieved Please explain.

*Figure 3 was referred by mistake to the CRA method instead to the* Method for Object-based Diagnostic Evaluation - MODE. *We added in the article the following correct explanation:*

Additional evidence on matching between our forecasts and satellite observations has been made by applying the Method for Object-based Diagnostic Evaluation - MODE (Davis et al., 2006a; 2006b; 2009) which is based on a fuzzy-logic algorithm and which has been originally developed to quantify the errors related to spatial patterns and location of precipitation which considers various attributes of rain patterns (e.g. orientation, rain area). Factors as the separation of the object (pattern) centroids, minimum edge separation between modeled and observed patterns, model/observed patterns orientation angles relative to the grid axis, the ratio of the areas of the two objects, and the fraction of area common to both objects. MODE is used here to indicate the level of matching between NL and IWPL for a selected day of 11 May 2010. Figure 3 shows that MODE has identified three precipitation objects: two (green and red colored) showing good matching, and one (blue) with no matching.

Page 8, Line 8: Anyhow, in order to predict IWC we need to incorporate predicted Nin into a cloud microphysics scheme, which is a future task of our project. Therefore, the comparison using a semi-quantitative approach is the only available at the current stage of the analysis. Why don't you incorporate the NMME microphysics scheme? Please show also the modeled IWC.

*See please above our reply to the General Comment of the Referee #1. Incorporating the NMME microphysics scheme will be the subject of the forthcoming Part 2 of the paper, as we have indicated above.*

Page 8, Line 14: Most of the ice, observed by the cloud radar below 4.0-4.5 km above ground level (AGL), is not predicted by the model. Is there any dust at these layers? If there is no dust in the model and your only IN source is dust, this could make sense.

*IN could be absent not only because dust missing but also because the other, thermodynamic conditions are not fulfilled. However, we assume that IN for lower mixed clouds is predicted because the DeMott scheme could not be extrapolated for T warmer than -36°C. See please also our detailed answer to the similar question of the Referee #2 on the same issue.*

Page 8, Line 23: Moreover, like for the case of May 2010, the model tends to under-predict the lowest ice water layers observed with the radar below 4.5 km AGL. Again it is a little confusing when you refer to IN and when you refer to ice water. Also to me it looks like there is no IN below 6km which means that the model fails to represent half of the clouds in this cross-section.

*We made corrections to avoid confusion when using model IN and ice water (following the Referee's comment.*

*We also included discussion addressed to the fact that the model failed to represent lower cold clouds.*

Inability of the model to predict $n_{IN}$ at lower elevations can be explained by the fact that the DeMott et al. (2015) parameterization is valid for temperatures in the interval (-20°C – -36°C). We extended this scheme to work in the interval (-5°C ; -20°C) as well but our experiments showed that lower mixed clouds could not be predicted. This result is consistent with the statement of DeMott et al. (2015) that the parameterization is weakly constrained at temperatures warmer than - 20°. As these authors also claimed, this is the temperature regime that may be dominated by organic ice nucleating particles such as ice nucleating bacteria, which is aerosol not included in our parameterizations.

Page 8, Line 27 and Figure 5 caption. Replace upper and lower panel with left and right

*We did it.*

Page 9, Line 18: On the contrary, in South Italy, the volcanic layer, observed at Potenza up to an altitude of about 8 km above sea level Please provide some evidence for this argument

The paragraph at page 9 has been modified by citing the sources that can provide the requested evidence for this argument. The information has been extended also to match the most recent published version of the lidar analysis of the observation collected in 2010 freely available in the relational database at [www.earlinet.org](http://www.earlinet.org). which extend the content of the local analysis performed at the CIAO EARLINET station in Potenza to the results achieved by the whole EARLINET at the European scale. This database contains all information about volcanic layers (base, top, center of mass) and correspondingly mean and integrated values. According to what discussed above, the new paragraph has been modified as follows:

In the period between 13-15 May 2010, both DREAM and back trajectories analysis showed that, while the transport of volcanic aerosol from Iceland (due to the Eruption of volcano Eyjafjalla 2010) was still ongoing, dust contribution was not negligible (Mona et al., 2012). In this period, the

volcanic aerosol was mainly 5 transported across the Atlantic Ocean, passing over Ireland and west UK, and then transported to the west off the Iberian Peninsula before reaching the Mediterranean Basin and Southern Italy. Satellite images and ground-based measurements confirmed the presence of volcanic particles in the corresponding regions (not shown). A detailed description of the volcanic layers as observed by EARLINET (European Aerosol Research Lidar NETwork) during this period is reported in Pappalardo et al., 2014. EARLINET volcanic dataset is freely available at www.earlinet.org (The EARLINET publishing group 2000-2010; (2014): EARLINET observations related to volcanic eruptions (2000-2010); World Data Center for Climate (WDCC). http://dx.doi.org/10.1594/WDCC/EN_VolcanicEruption_2000-2010).Moreover a devoted relational database freely avialable at www.earlinet.org contains all information about volcanic layers (base, top, center of mass) and correspondingly mean and integrated values.

The Iberian Peninsula, France and South Italy were the regions more significantly affected by the presence of volcanic aerosol (sulphate and small ash) during the considered period. For the purpose of our modelling study this might induce an underestimation of the IN (since IN due to dust only is modeled) in the above mentioned regions and can be responsible of part of the discrepancies between modeled IN and IWP provided by SEVIRI. This is particularly true for Iberian Peninsula where volcanic aerosol concentrations were quite relevant. The comparison of model predicted IN and SEVIRI IWC on 13 May shows differences that might be correlated to a larger availability of IN of volcanic origin.

On the contrary, in South Italy, the volcanic layer, observed at Potenza up to an altitude up to 15.8 km above sea level, did not enhance the formation of cold clouds due to unfavourable dry conditions in the free troposphere; this is also confirmed by the Potenza cloud radar which did not observed? clouds for the whole day (Figure 4). The absence of cold clouds over most of South Italy, including Potenza region, is also shown by the IWC reported for 13 May in Figure 1.

Page 9 Line 20: did not observed? Typo - observe

*corrected*

Page 9, Line 29 : The model has been validated .Avoid the use of the term validation (here and elsewhere in the text) since you are only referring to specific case studies. A validation process would require much more comparisons with observations and for a much longer time period until the model could be verified to produce validated products.

*accepted and reformulated*

Page 9, Line 32: wormer negative temperatures Typo - warmer

*corrected*

Page 10, Line 6: What do you mean by "unified modelling system"

*The word 'unified' is redundant and we removed it.*

Throughout the text Please check again the text for grammar and spelling and provide an improved manuscript.

*We made effort and improved the language with the assistance of a native English colleague.*

**References**

*Burton, S. P., Ferrare, R. A., Vaughan, M. A., Omar, A. H., Rogers, R. R., Hostetler, C. A., and Hair, J. W.: Aerosol classification from airborne HSRL and comparisons with the CALIPSO vertical feature mask, Atmos. Meas. Tech., 6, 1397-1412, doi:10.5194/amt-6-1397-2013, 2013.*

*Groß, S., Freudenthaler, V., Schepanski, K., Toledano, C., Schäfler, A., Ansmann, A., and Weinzierl, B.: Optical properties of long-range transported Saharan dust over Barbados as measured by dual-wavelength depolarization Raman lidar measurements, Atmos. Chem. Phys., 15, 11067-11080, doi:10.5194/acp-15-11067-2015, 2015.*

*James D. Atkinson, Benjamin J. Murray, Matthew T. Woodhouse, Thomas F. Whale, Kelly J.Baustian, Kenneth S. Carslaw, Steven Dobbie, Daniel O'Sullivan & Tamsin L. Malkin, The importance of feldspar for ice nucleation by mineral dust in mixed-phase clouds, Nature 498, 355–358 doi:10.1038/nature12278, 2013*

*K. Zhang, D. O'Donnell, J. Kazil, P. Stier, S. Kinne, U. Lohmann, S.Ferrachat, B. Croft, J. Quaas,H. Wan, S. Rast, and J.Feichter, The global aerosol-climate model ECHAM-HAM, version 2:sensitivity to improvements in process repre-sentations, Atmos. Chem. Phys., 12, 8911–8949, 2012 www.atmos-chem-phys.net/12/8911/2012/doi:10.5194/acp-12-8911-2012*

*X. Liu, X. Shi, K.Zhang, E. J. Jensen, A. Gettelman, D. Barahona, A.Nenes, and P. Lawson, SensitivitystudiesofdusticenucleieffectoncirruscloudswiththeCommunity Atmosphere Model CAM5, Atmos. Chem. Phys., 12, 12061–12079, 2012 www.atmos-chem-phys.net/12/12061/2012/ doi:10.5194/acp-12-12061-2012*

*Thompson, G and T.Eidhammer, 2014: A Study of Aerosol Impacts on Clouds and Precipitation Development in a Large Winter Cyclone. J. Atmos. Sci., 71, 3636–3658.*

*Sessions, W. R., Reid, J. S., Benedetti, A., Colarco, P. R., da Silva, A., Lu, S., Sekiyama, T., Tanaka, T. Y., Baldasano, J. M., Basart, S., Brooks, M. E., Eck, T. F., Iredell, M., Hansen, J. A., Jorba, O. C., Juang, H.-M. H., Lynch, P., Morcrette, J.-J., Moorthi, S., Mulcahy, J., Pradhan, Y., Razinger, M., Sampson, C. B., Wang, J., and Westphal, D. L.: Development towards a global operational aerosol consensus: basic climatological characteristics of the International Cooperative for Aerosol Prediction Multi-Model Ensemble (ICAP-MME), Atmos. Chem. Phys., 15, 335-362, doi:10.5194/acp-15-335-2015, 2015.*

*Ferrier, B. S., Jin, Y., Lin, Y., Black, T., Rogers, E., and DiMego, G., 2002: Implementation of a new grid-scale cloud and precipitation scheme in the NCEP Eta Model. Proc. 15th Conf. on Numerical Weather Prediction, San Antonio, TX, Amer. Meteor. Soc., pp. 280–283.*

*Janjic, Z. I., 2000: Comments on "Development and Evaluation of a Convection Scheme for Use in Climate Models". Journal of the Atmospheric Sciences, 57, 3686–3686.*

*Janjic, Z. I., 1994: The step-mountain eta coordinate model: further developments of the convection, viscous sublayer and turbulence closure schemes. Monthly Weather Review, Vol. **122**, 927-945.*

---

## Author Comment (AC2) · 24 Aug 2016

Page 2 line 28: To our knowledge, this is the first time that all ingredients needed for cold cloud formation by dust are predicted in operational forecasting mode within one modeling system. Please give more evidence.

*Please see our reply to the General Comment of the Referee #1*

Page 4 line 25: the spread of errors in predicting IN concentrations at a given temperature has been reduced from a factor of 1000 to 10. Please give some evidence or support for this conclusion.

*This statement is based on results shown Fig 3 in DeMott et al (2010); see the reference below. The spread of predicted/observed $n_{IN}$ points range from 0.001 to 10 ($L^{-1}$) in the older approach of Meyers et al.(1992); in the DeMott et al (2010) article, it ranges approximately from 0.001 to 0.1 ($L^{-1}$). We introduced in the article corresponding reference of DeMott et al, 1010.*

Page 4 line 29: Why do you choose -5C, since the underlying measurements were taken at temperatures lower than -9C. Moreover, you set the temperatures for C1 warmer clouds range -36C to -10C (page 4 line 17 ). Please give more discussion.
and
Page 8 line 14-15: Why the model can't predict the ice below 4-4.5km while the cloud radar can detect? Due to the temperatures you set in section 2.3 (-10 – -36C) , the vertical distribution of dust aerosols, or any other reason? You should give more discussion.

*Although the DeMott et al. (2015) parameterization is for temperatures (-20°C – -36°C) we extrapolated their scheme to work in the interval (-5°C – -20°C)as well, with intention to include prediction of the occurrence of warmer mixed clouds. Our experiments however showed that the scheme could not predict such clouds. Probable reason for that is that the parameterization is weakly constrained at temperatures at temperatures warmer than -20°C as stated by DeMott et al. (2015). As these authors claimed, this is the temperature regime that may be dominated by organic ice nucleating particles such as ice nucleating bacteria. In the article we added the following text:*

Inability of the model to predict $n_{IN}$ at lower elevations can be explained by the fact that the DeMott et al. (2015) parameterization is valid for temperatures in the interval (-20°C – -36°C). We extended this scheme to work in the interval (-5°C ; -20°C) as well but our experiments showed that lower mixed clouds could not be predicted. This result is consistent with the statement of DeMott et al. (2015) that the parameterization is weakly constrained at temperatures warmer than -20°. As these authors also claimed, this is the temperature regime that may be dominated by organic ice nucleating particles such as ice nucleating bacteria, which is aerosol not included in our parameterizations.

Page 5 line 9: Dust is ice nucleation active surface linked to dust concentration. As we know that dust aerosols lifted to the mid and upper troposphere can serve as ice nuclei, here you use the surface value of dust. Will it affect the model results?

*We wish to clarify that surface is not addressed to the dust concentration on surface but to the ice-active surface site density $S_{dust}$ [$m^{-2}$] (see also the definition of $S_{dust}$ in e.g. Connolly et al., 2009; Niemand et al., 2012). $S_{dust}$ describes the ability of a dust particle to freeze the cloud water. We added a clarification in the text.*

Page 5 line 12-13: Please give some related evidence.

*In our IN modelling, we transit from DeMott et al., (2015) to Steinke et al., (2015) parameterization at T= -36°C. Although one might expect some discontinuity to occur at this transitional temperature threshold, in practice a rather continuous behavior of $n_{IN}$ at this boundary has been achieved, thus not additional smoothing has been required. The following clarification has been introduced in the article in response to the Referee's request:*

Although based on two different parameterizations, the resulting $n_{IN}$ has a smooth transition across the temperature boundary of -36°C between DeMott et al. (2015) and Steinke et al. (2015) schemes. At this transitional temperature, we have not applied any mathematical smoothing.

Page 5: paragraph 1 on section 3, please give details description for the ground observe instruments.

*The paragraph 1 on section 3 has been extended in the new version of the manuscript to provide a more detailed description of the ground based instruments employed in the presented study. The new paragraph is reported below for your convenience:*

[revised manuscript text omitted]

Page 8 line 27: The position for the pictures in Fig.5 should be left and right.

*Corrected*

Page 9 line 20: there is a redundant question mark.

*Corrected*

Figure 1: The color bar and coordinate are unclear. The compared results for the second case should also be given and discussed.

*We replaced the figure with its vector image format in which the color bar can be checked by enlarging the image.*

*Comparison against SEVIRI for the second case: We did not include comparison model NL against SEVIRI IWPL because unfortunately EUMETSAT CN SAF products are available for the period (2009-01-01 - 2012-02-29) and do not cover September 2012. See:* *https://wui.cmsaf.eu/safira/action/viewProduktDetails?id=11467_14338_16270_16283_16906*

Figure 3: Please give the meaning for each color and the title for x-y coordinate.

*We added the explanation for the meaning of each color and the meaning of x-y coordinate.*

Figure 5: For the first case, the mean values of IWPL are mainly greater than NL. However, for the second case, the mean values of IWPL are mainly less than NL. Please give more discussion.

*IWPL vs. NL: This is due to our mistake in inserting wrong image on the right, see graphs below which are now correctly included in the article.*

[Figure]

[Figure]

There are some research discuss dust aerosols effect on clouds and precipitation, please discuss more about the relationship between dust and clouds in Section 1.

*The following was added in reply to the Referee's request.*

Another study indicates that desert dust has the ability to glaciate the top of developing convective clouds, creating ice precipitation instead of suppressing warm rain; also dust invigoration effect would enhance precipitation ( Rosenfeld et al., 2008). On the other hand, Teller et al. (2012) conclude from their modelling study that the presence of mineral dust had a much smaller effect on the total precipitation than on its spatial distribution, which indicates that quantification of dust effects to precipitation is still uncertain because dust could modify cloud properties in many complex ways (Huang et al., 2014); therefore impacts of dust on cloud processes requires further research.

---

## Author Response (AR1)

**General Comment:**

My main concern is that at this stage the authors compare model ice nuclei with observations of ice water path which are not directly comparable. It would be more appropriate to calculate the corresponding modeled cloud properties and use them for model evaluation. Model results with the old version of the model should be also presented for comparison.

We agree, it would be more appropriate to compare the model against the observed cloud ice. However, in the current stage of our work, we do not predict the cloud ice but calculate the ice nuclei concentration  $(n_{IN})$  and therefore compared  $n_{IN}$  against the observed ice water. Although the two variables are not directly comparable as the Referee #1 correctly noticed, they are linked through the continuity equation for cloud ice and we assumed the two variables should be correlated. Our results indeed confirm that  $n_{IN}$  in general well compares against the observed cloud ice water.

In our next paper which will represent as continuation of the current study, we will use  $n_{IN}$  as a prognostic input in the cloud microphysics scheme of our dust-atmosphere modelling system. Based on above considerations, we propose to the Editor replacement of the current manuscript title with: 'Cloud ice caused by atmospheric mineral dust - Part 1: Parameterization of ice nuclei concentration in the NMME-DREAM model'. This also takes into account the Referee's suggestion in his/her first Specific Comment to change the article title. The second part of the current in that case will be a paper titled as: 'Cloud ice caused by atmospheric mineral dust - Part 2: Parameterization of ice water in the NMME-DREAM model'.

**Specific Comments:**

Page2, Line 28: To our knowledge, this is the first time that all ingredients needed for cold cloud formation by dust are predicted in the operational forecasting mode within one modeling system. Please provide more support for this statement since a variety of coupled dust-ice models seem to be already available (e.g. Zhang et al., 2012; Liu et al., 2012; Atkinson et al., 2013).

This is true there are numerous coupled dust-ice models. Articles the suggested by the Referee #1 (Zhang et al., 2012; Liu et al., 2012; Atkinson et al., 2013) are focused on studying nucleation effects in general, rather than on using dust-ice parameterization to improve numerical weather prediction. For example,  $n_{IN}$  is not online prognostic variable in none of the operational dust models within the largest two international dust forecasting the WMO Sand and Dust Warning and Assessment projects: System https://www.wmo.int/pages/prog/arep/wwrp/new/Sand and Dust Storm.html; and the ICAP Multi-Model Ensemble (ICAP-MME) http://icap.atmos.und.edu/; Sessions et al, 2015). Differently from those dust models, we perform prediction of  $n_{IN}$  at every model time step to be used as input to a microphysics scheme.

We added in the text the following clarification:

... Such new parameter will be used in our future study as an input to a microphysics scheme, expecting improve the operational prediction of cold clouds and associated precipitation. Currently,  $n_{IN}$  is not used as online prognostic variable in eider of the operational dust models of two largest

international dust forecasting networks: in the WMO Sand and Dust Warning and Assessment System (SDS-WAS) (://www.wmo.int/pages/prog/arep/wwrp/new/Sand\_and\_Dust\_Storm.html) and in the ICAP Multi-Model Ensemble (ICAP-MME) (http://icap.atmos.und.edu/). Unlike dust models of these networks, our modelling system predicts  $n_{IN}$  at every model time step which will be used as input to a microphysics scheme in the study of our forthcoming paper...

Page 4, Line 14 : In this study, dust concentration, atmospheric temperature and moisture as predicted by the atmospheric component of the coupled model are used to calculate. The parameterization consists of two parts applied to warmer and colder glaciated clouds. The vertical wind component is a crucial parameter for CCN/IN activation processes. Do you consider w in your calculations?

We wanted to implement the most recent parameterizations available in the community for dust-induced IN (DeMott et al., 2015; Steinke et al., 2015). These schemes require temperature, relative humidity and dust concentration as input parameters, but not vertical velocity. We added the following text:

The schemes of DeMott et al. and Steinke et al. require temperature, relative humidity and dust concentration as input parameters, but not vertical velocity as it is used in some other microphysical schemes (e.g. Wang et al, 2014).

Page 5, Line 20: to identify the different aerosol types (Papagiannopoulos et al., 2015) taking advantage of the large number optical properties they are able to provide, i.e. lidar ratio at two wavelengths, the Angstrom exponent, the backscatter-related Angstrom exponent, and linear particle depolarization ratio. This aerosol typing capability allows to classify the aerosol type acting Nin, and especially to separate mineral dust from other types of aerosol Please add Papagiannopoulos et al., 2015 in your Reference list. Also check carefully your references and edit your list in ACP format.

The reference by Papagiannopoulos et al. has been replaced by the following two as more appropriate:

Groß, S., Freudenthaler, V., Schepanski, K., Toledano, C., Schäfler, A., Ansmann, A., and Weinzierl, B.: Optical properties of long-range transported Saharan dust over Barbados as measured by dual-wavelength depolarization Raman lidar measurements, Atmos. Chem. Phys., 15, 11067-11080, doi:10.5194/acp-15-11067-2015, 2015.

Burton, S. P., Ferrare, R. A., Vaughan, M. A., Omar, A. H., Rogers, R. R., Hostetler, C. A., and Hair, J. W.: Aerosol classification from airborne HSRL and comparisons with the CALIPSO vertical feature mask, Atmos. Meas. Tech., 6, 1397-1412, doi:10.5194/amt-6-1397-2013, 2013.

Page 6, Line 15: The model resolution has been set to 25km in the horizontal. Could you please justify how you resolve cloud-scale features at this resolution?

We added the following text as a clarification:

At the horizontal model resolution used in our study (which relates to the hydrostatic type of thermodynamics), clouds are resolved by the following schemes: the parameterization of grid-scale clouds and microphysics (Ferrier et al., 2002); and the parameterization of convection clouds (Janjić, 1994, 2000).

Page 7, Line 4. On the other hand, a visual inspection shows considerable similarity between NL and the IWPL patterns (columns (B) and (C)) with respect to their shapes and locations. These two quantities are not directly comparable. Could you please show what is the NMME predicted IWP? Also show the difference between the control run (without IN parameterization) and the new run.

See please above our reply to the General Comment of the Referee #1

Figure 2. If I interpret correct the plots in Figure 2, it seems that the model predicts IN even at areas without dust. If your only aerosol source is dust (Eq.1, Eq.2) could you please explain more on this?

We think the Referee #1 is reporting to Figure 1, not to Figure 2.

Figure 1 shows maps of dust C and IN integrated vertically (columns A and B respectively). The maps for same valid times indeed do not fully match because C and IN are not linearly proportional (Eq.1, Eq.2), neither their loads. However, according to Eq.1 and Eq.2 even for small C concentration there is some  $n_{IN}$  if thermodynamic conditions permit it. With the used color palette scales in maps, load of small dust concentrations cannot be shown even if  $n_{IN}$  is displayed.

Page 7, Line 29: The forecasts are translated horizontally over the observations until the minimum squared error (MSE) is achieved Please explain.

Figure 3 was referred by mistake to the CRA method instead to the Method for Object-based Diagnostic Evaluation - MODE. We added in the article the following correct explanation:

Additional evidence on matching between our forecasts and satellite observations has been made by applying the Method for Object-based Diagnostic Evaluation - MODE (Davis et al., 2006a; 2006b; 2009) which is based on a fuzzy-logic algorithm and which has been originally developed to quantify the errors related to spatial patterns and location of precipitation which considers various attributes of rain patterns (e.g. orientation, rain area). Factors as the separation of the object (pattern) centroids, minimum edge separation between modeled and observed patterns, model/observed patterns orientation angles relative to the grid axis, the ratio of the areas of the two objects, and the fraction of area common to both objects. MODE is used here to indicate the level of matching between NL and IWPL for a selected day of 11 May 2010. Figure 3 shows that MODE has identified three precipitation objects: two (green and red colored) showing good matching, and one (blue) with no matching.

Page 8, Line 8: Anyhow, in order to predict IWC we need to incorporate predicted Nin into a cloud microphysics scheme, which is a future task of our project. Therefore, the comparison using a semi-quantitative approach is the only available at the current stage of the analysis. Why don't you incorporate the NMME microphysics scheme? Please show also the modeled IWC.

See please above our reply to the General Comment of the Referee #1. Incorporating the NMME microphysics scheme will be the subject of the forthcoming Part 2 of the paper, as we have indicated above.

Page 8, Line 14: Most of the ice, observed by the cloud radar below 4.0-4.5 km above ground level (AGL), is not predicted by the model. Is there any dust at these layers? If there is no dust in the model and your only IN source is dust, this could make sense.

IN could be absent not only because dust missing but also because the other, thermodynamic conditions are not fulfilled. However, we assume that IN for lower mixed clouds is predicted because the DeMott scheme could not be extrapolated for T warmer than -36°C. See please also our detailed answer to the similar question of the Referee #2 on the same issue.

Page 8, Line 23: Moreover, like for the case of May 2010, the model tends to under-predict the lowest ice water layers observed with the radar below 4.5 km AGL. Again it is a little confusing when you refer to IN and when you refer to ice water. Also to me it looks like there is no IN below 6km which means that the model fails to represent half of the clouds in this cross-section.

We made corrections to avoid confusion when using model IN and ice water (following the Referee's comment.

We also included discussion addressed to the fact that the model failed to represent lower cold clouds.

Inability of the model to predict  $n_{IN}$  at lower elevations can be explained by the fact that the DeMott et al. (2015) parameterization is valid for temperatures in the interval (-20°C – -36°C). We extended this scheme to work in the interval (-5°C; -20°C) as well but our experiments showed that lower mixed clouds could not be predicted. This result is consistent with the statement of DeMott et al. (2015) that the parameterization is weakly constrained at temperatures warmer than -20°. As these authors also claimed, this is the temperature regime that may be dominated by organic ice nucleating particles such as ice nucleating bacteria, which is aerosol not included in our parameterizations.

Page 8, Line 27 and Figure 5 caption. Replace upper and lower panel with left and right *We did it.*

Page 9, Line 18: On the contrary, in South Italy, the volcanic layer, observed at Potenza up to an altitude of about 8 km above sea level Please provide some evidence for this argument

The paragraph at page 9 has been modified by citing the sources that can provide the requested evidence for this argument. The information has been extended also to match the most recent published version of the lidar analysis of the observation collected in 2010 freely available in the relational database at www.earlinet.org. which extend the content of the local analysis performed at the CIAO EARLINET station in Potenza to the results achieved by the whole EARLINET at the European scale. This database contains all information about volcanic layers (base, top, center of mass) and correspondingly mean and integrated values. According to what discussed above, the new paragraph has been modified as follows:

In the period between 13-15 May 2010, both DREAM and back trajectories analysis showed that, while the transport of volcanic aerosol from Iceland (due to the Eruption of volcano Eyjafjalla 2010) was still ongoing, dust contribution was not negligible (Mona et al., 2012). In this period, the

volcanic aerosol was mainly 5 transported across the Atlantic Ocean, passing over Ireland and west UK, and then transported to the west off the Iberian Peninsula before reaching the Mediterranean Basin and Southern Italy. Satellite images and ground-based measurements confirmed the presence of volcanic particles in the corresponding regions (not shown). A detailed description of the volcanic layers as observed by EARLINET (European Aerosol Research Lidar NETwork) during this period is reported in Pappalardo et al., 2014. EARLINET volcanic dataset is freely available at www.earlinet.org (The EARLINET publishing group 2000-2010; (2014): EARLINET observations related to volcanic eruptions (2000-2010);World Data Center for Climate (WDCC). http://dx.doi.org/10.1594/WDCC/EN VolcanicEruption 2000-2010).Moreover a devoted relational database freely avialable at www.earlinet.org contains all information about volcanic layers (base, top, center of mass) and correspondingly mean and integrated values.

The Iberian Peninsula, France and South Italy were the regions more significantly affected by the presence of volcanic aerosol (sulphate and small ash) during the considered period. For the purpose of our modelling study this might induce an underestimation of the IN (since IN due to dust only is modeled) in the above mentioned regions and can be responsible of part of the discrepancies between modeled IN and IWP provided by SEVIRI. This is particularly true for Iberian Peninsula where volcanic aerosol concentrations were quite relevant. The comparison of model predicted IN and SEVIRI IWC on 13 May shows differences that might be correlated to a larger availability of IN of volcanic origin.

On the contrary, in South Italy, the volcanic layer, observed at Potenza up to an altitude up to 15.8 km above sea level, did not enhance the formation of cold clouds due to unfavourable dry conditions in the free troposphere; this is also confirmed by the Potenza cloud radar which did not observed? clouds for the whole day (Figure 4). The absence of cold clouds over most of South Italy, including Potenza region, is also shown by the IWC reported for 13 May in Figure 1.

Page 9 Line 20: did not observed? Typo - observe

corrected

Page 9, Line 29 : The model has been validated .Avoid the use of the term validation (here and elsewhere in the text) since you are only referring to specific case studies. A validation process would require much more comparisons with observations and for a much longer time period until the model could be verified to produce validated products.

accepted and reformulated

Page 9, Line 32: wormer negative temperatures Typo - warmer

corrected

Page 10, Line 6: What do you mean by "unified modelling system"

The word 'unified' is redundant and we removed it.

Throughout the text Please check again the text for grammar and spelling and provide an improved manuscript.

We made effort and improved the language with the assistance of a native English colleague.

Page 5: paragraph 1 on section 3, please give details description for the ground observe instruments.

The paragraph 1 on section 3 has been extended in the new version of the manuscript to provide a more detailed description of the ground based instruments employed in the presented study. The new paragraph is reported below for your convenience:

[revised manuscript text omitted]

Page 8 line 27: The position for the pictures in Fig.5 should be left and right.

Corrected

Page 9 line 20: there is a redundant question mark.

Corrected

Figure 1: The color bar and coordinate are unclear. The compared results for the second case should also be given and discussed.

We replaced the figure with its vector image format in which the color bar can be checked by enlarging the image.

Comparison against SEVIRI for the second case: We did not include comparison model NL against SEVIRI IWPL because unfortunately EUMETSAT CN SAF products are available for the period (2009-01-01 - 2012-02-29) and do not cover September 2012. See: https://wui.cmsaf.eu/safira/action/viewProduktDetails?id=11467\_14338\_16270\_16283\_16906

Figure 3: Please give the meaning for each color and the title for x-y coordinate.

We added the explanation for the meaning of each color and the meaning of x-y coordinate.

Figure 5: For the first case, the mean values of IWPL are mainly greater than NL. However, for the second case, the mean values of IWPL are mainly less than NL. Please give more discussion.

*IWPL vs. NL: This is due to our mistake in inserting wrong image on the right, see graphs below which are now correctly included in the article.*

There are some research discuss dust aerosols effect on clouds and precipitation, please discuss more about the relationship between dust and clouds in Section 1.

The following was added in reply to the Referee's request.

Another study indicates that desert dust has the ability to glaciate the top of developing convective clouds, creating ice precipitation instead of suppressing warm rain; also dust invigoration effect would enhance precipitation (Rosenfeld et al., 2008). On the other hand, Teller et al. (2012) conclude from their modelling study that the presence of mineral dust had a much smaller effect on the total precipitation than on its spatial distribution, which indicates that quantification of dust effects to precipitation is still uncertain because dust could modify cloud properties in many complex ways (Huang et al., 2014); therefore impacts of dust on cloud processes requires further research.

**20 Keywords**

5

ice nucleation, model parameterization, dust aerosol

**Abstract**

Dust aerosols are very efficient ice nuclei, important for heterogeneous cloud glaciation even in regions distant from desert sources. A new generation of ice nucleation parameterizations, including dust as ice nucleation agent, opens the way towards a more accurate treatment of cold cloud formation in atmospheric models. Using such parameterizations, we have developed a regional dust-atmospheric modelling system capable of predicting to predict in real-time, conditions, dust-induced ice nucleation. We executed the model with the added ice nucleation component over the Mediterranean region, exposed to moderate Saharan dust transport, over two periods lasting 15 and 9 days, respectively. The modelModel results were comparedare validated against satellite and ground-based cloud-ice-related measurements, provided by SEVIRI (Spinning Enhanced Visible and InfraRed Imager) and by the CNR-IMAA Atmospheric Observatory CIAO in Potenza, South Italy. The predictedPredieted ice nuclei concentration showed ashows reasonable level of agreement when compared against the observed spatial and temporal patterns of cloud ice water. The developed methodology permits thete, use of

| Formatted: English (United Kingdom)                                                                                     |
|-------------------------------------------------------------------------------------------------------------------------|
| Formatted: Authors, Adjust space
between Latin and Asian text, Adjust
space between Asian text and numbers |
| Formatted: English (United Kingdom)                                                                                     |
| Formatted: Font: (Default) Times New
Roman, 12 pt, Not Bold, English (United
Kingdom)                             |
| Formatted: Font: 11 pt                                                                                                  |
| Formatted: Font: 11 pt                                                                                                  |

[revised manuscript text omitted]

variety of geographic locations over a period longer than a decade, demonstrating that there is a correlation between the observed  $p_{dN_A}$  and the dust number concentrations of particles larger than 0.25µm in radius. In DeMott et al. (2015), C = =3 is chosen as a calibration factor to adjust the scheme to dust measurements. Despite The parameterization is extrapolated down to -5°C despite the fact that validity of the scheme is forunderlying measurements were only taken at temperatures colder than -20°C we extrapolated its application down to -5°C in order to test the model if it can predict the occurrence of lower mixed clouds for the temperatures range being out of the validity of the parameterization scheme.than -9°C.

For temperatures ranging in the interval (-55°C; -36°C), we have implemented the Steinke et al. (2015) parameterization for the deposition ice nucleation based on the ice nucleation active surface site approach in which  $n_{IN}$  is a function of temperature, humidity and the aerosol surface area concentration. In the deposition nucleation, water 10 vapourvapor is directly transformed into ice at the particle's surface, occurring at the time of or shortly after the water condensation on the particle, which acts at the same time as a condensation and freezing nucleus. <del>nuclei. For deposition</del> <del>nucleation, water vapor is directly transformed into ice at the particle's surface.</del> Steinke et al. (2015) calculate the number

concentration of ice nuclei due to deposition freezing as:

 $n_{IN} = pS_{dust} \exp[-q(T - 273.16) + (rRH_{ice} - 100)]$

15

20

25

30

5

here  $n_{IN}$  is here  $n_{IN}$  is here  $n_{IN}$  is the number concentration of ice nuclei  $[cm_{3}^{-3}]$ ;  $S_{dust}$  is the the *is*-ice-nucleation-active surface site density  $[m_{2}^{-2}]$  linked to dust concentration (Niemand et al., 2012) describing the efficiency of a dust particle to freeze the cloud water.  $\frac{1}{2} p = 188 \times 10^{5}; q = -1.0815; r = -0.815; T$  is temperature in degrees Celsius;  $RH_{ice}$  is relative humidity with respect to ice. In our experiments,  $RH_{ice}$  is prespecified is pre-specified to the value of 110%.

Although based on two different parameterizations, the resulting  $n_{IN}$  has a smooth transition across the temperature boundary of -36°C between the-DeMott et al. (2015) and Steinke et al. (2015) scheme, as our model results shown later demonstrate. Therefore, there was no need to numerically smooth  $n_{IN}$  of the two-schemes. At this transitional temperature, we have not applied any mathematical smoothing. to secure appropriate matching.

[revised manuscript text omitted]

To complement the Potenza in-situ profiling observations and to examine how the model predicts horizontal distribution of cold clouds, the MSG/SEVIRI ice water path satellite observations wereare used. SEVIRI (the Spinning Formatted: English (United Kingdom) Formatted: English (United Kingdom) Formatted: English (United Kingdom)

Enhanced Visible and InfraRed Imager), as a geostationary passive imager, is on board of the Meteosat Second Generation (MSG) systems. The high High SEVIRI spatial and temporal resolution (~4km and 15min, respectively), provides, among other advantages, provides, high-quality products. The inputsInput to the retrieval schemes were inter-calibrated effective radiances of Meteosat-8 and 9. In our study, the daily averages of the retrieved ice water path of the SEVIRI cloud property dataset (CLAAS) wereare used (Stengel et al., 2013a; Stengel et al., 2013b) to comparevalidate the model results against there observations:on the regional scale:

 $IWP = \frac{2}{3}r_Ir_{eff}\tau$

[revised manuscript text omitted]

level (AGL), is not predicted by the model. This is particularly evident on 6 May when only-ice cloud layer below 3 km AGL wasis observed only by the radar, and conditions for ice occurrence werebut completely missed by the model.

On 22-30 September 2012 the model wasis able to indicate ofeatch the deep ice layers observed on 25-27-September 2012 between about 5 and 12 km AGL (-10°C and -60°C) and it wasis able to partially predict a part of the thinner layers observed after 27 September above 7 km AGL (<-25°C). The model wasis also able to well-predict well the occurrence of cirrus clouds observed by the cloud radar on 29 September in the range between 6 and 12 km. It is also worth to mention that the co-located and simultaneous Raman lidar measurements (not reported) showed some high optically thin cloudiness not detected by the radar because of its limited sensitivity to thin clouds at that height <del>levels</del> (Borg et al., 2013). Formatted: English (United Kingdom) Formatted: English (United Kingdom) Formatted: English (United Kingdom) Formatted: Indent: First line: 1,27 cm

In particular, this is the  $n_{IN}$  case of the layers predicted by the model in the second half of 27 September and on 28 September are in the range between 9 and 12 km. However, as in Moreover, like for the case of May 2010, the model underpredicted  $n_{IN}$  fortends to underpredict the lowest ice water layers observed with the radar below 4.5 km. AGL.

Inability of the model to predict  $n_{IN}$  at lower elevations can be explained by the fact that the DeMott et al. (2015) parameterization is valid for temperatures in the interval (-20°C – -36°C). We extended this scheme to work in the interval (-5°C ; -20°C) as well but our experiments showed that lower mixed clouds could not be predicted. This result is consistent with the statement of DeMott et al. (2015) that the parameterization is weakly constrained at temperatures warmer than -20°. As these authors also claimed, this is the temperature regime that may be dominated by organic ice nucleating particles such as ice nucleating bacteria, which is aerosol not included in our parameterizations.

10

In Figure 5 we also report the comparison of IWPL and NL over Potenza calculated every three hours, in the period from 1 to 15 May 2010 (left(upper panel) and from 22 to 30 September 2012 (right(lower panel)). The outcome of the comparison confirms the good performance of the model in the prediction of  $n_{IN}$  of the ice clouds over the whole atmospheric column.

15

The correlation between the IWPL and NL retrieved using the ground based measurements, merging the datasets from both the-selected cases studies of 1-15 May 2010 and 22-30 September 2012, is shown in Figure 6. The linearLinear correlation made considering the daily averagesaveraged for both the quantities provides a regression coefficient of R=0.83. The scatter plot shows a large variability in the values corresponding to the higher values of the IWP and to the higher values of IL. Therefore, for optically thinner ice clouds, IL linearly increases with IWPL. For larger IWPL values, the two variables are less correlated and the second or higher order polynomial fitting could compromise the linear relationship.

20

25

30

In the period between 13-15 May 2010, 2010, both DREAM and back trajectories analysis showed that, while the transport of volcanic aerosol from Iceland (due to the Eruption of volcano Eyjafjalla 2010) was still ongoing, dust contribution was not negligible (Mona et al., 2012). In this period, the volcanic aerosol was mainly 5 transported across the Atlantic Ocean, passing over Ireland and west UK, and then transported to the west off the Iberian Peninsula before reaching the Mediterranean Basin and Southern Italy. Satellite images and ground-based measurements confirmed the presence of volcanic particles in the corresponding regions (not shown). The analysis of multi wavelength Raman lidar mea permitted a detailed aerosol typing at the different altitude levels over Europe. A detailed description of the volcanic layers as observedlidar measurements performed, by EARLINET (European Aerosol Research Lidar NETwork)LIdarNETwork) over Europe, during this period is was 
[revised manuscript text omitted]